# Primordial germ cell DNA demethylation and development require DNA translesion synthesis

Pranay Shah [1] ✉, Ross Hill [1], Camille Dion[2,3], Stephen J. Clark[4,5], Abdulkadir Abakir[4,5], Jeroen Willems [6], Mark J. Arends [7], Juan I. Garaycoechea [6], Harry G. Leitch [2,3], Wolf Reik [4,5] & Gerry P. Crossan [1] ✉

Mutations in DNA damage response (DDR) factors are associated with human infertility, which affects up to 15% of the population. The DDR is required during germ cell development and meiosis. One pathway implicated in human fertility is DNA translesion synthesis (TLS), which allows replication impediments to be bypassed. We find that TLS is essential for pre-meiotic germ cell development in the embryo. Loss of the central TLS component, REV1, significantly inhibits the induction of human PGC-like cells (hPGCLCs). This is recapitulated in mice, where deficiencies in TLS initiation ($Rev1^{-/-}$ or $Pcna^{K164R/K164R}$) or extension ($Rev7^{-/-}$) result in a > 150-fold reduction in the number of primordial germ cells (PGCs) and complete sterility. In contrast, the absence of TLS does not impact the growth, function, or homeostasis of somatic tissues. Surprisingly, we find a complete failure in both activation of the germ cell transcriptional program and in DNA demethylation, a critical step in germline epigenetic reprogramming. Our findings show that for normal fertility, DNA repair is required not only for meiotic recombination but for progression through the earliest stages of germ cell development in mammals.

The development of germ cells and their differentiation into gametes is crucial for the faithful transmission of both genetic and epigenetic information to the next generation. Primordial germ cells (PGCs) are the first germ cells to emerge in the post-implantation embryo. In mice, as few as 3−5 founder PGCs are specified at E6.0-6.5 and these undergo extensive proliferation, increasing to around 20,000 by E12.5[1–3]. To develop into functional gametes, PGCs must undergo a unique developmental program, repressing somatic genes and activating pluripotency and germ-cell-specific factors[4,5]. This entails extensive epigenetic reprogramming resulting in altered histone modifications and DNA demethylation which facilitates the erasure of genomic imprints, and reactivation of the inactive X-chromosome[6,7].

DNA demethylation has been proposed to occur by multiple mechanisms[8]. A prevalent model describes a two-step process involving a passive demethylation phase in which DNA methylation is diluted by DNA replication ('reprogramming step 1') followed by active, enzymatic DNA demethylation that occurs upon colonization of the embryonic gonads ('reprogramming step 2')[9–14]. Thus, PGC development is highly dependent on DNA replication, both for lineage expansion and for epigenetic reprogramming.

The replication of DNA can be hindered by various obstacles, such as chemical damage to the DNA molecule or DNA secondary structures[15]. The failure to resolve these impediments can have catastrophic consequences for a cell. Incomplete or under-replication of

[1]MRC Laboratory of Molecular Biology, Cambridge CB2 0QH, UK. [2]MRC Laboratory of Medical Sciences, London W12 0HS, UK. [3]Institute of Clinical Sciences, Faculty of Medicine, Imperial College London, London W12 0HS, UK. [4]Altos Labs, Cambridge, UK. [5]Epigenetics Programme, The Babraham Institute, Cambridge CB22 3AT, UK. [6]Hubrecht Institute, Royal Netherlands Academy of Arts and Sciences (KNAW), Utrecht, The Netherlands. [7]Cancer Research UK Edinburgh Centre, Edinburgh, UK. ✉e-mail: pranayshah96@gmail.com; gcrossan@mrc-lmb.cam.ac.uk

the genome can block cell cycle progression, either directly or by activation of the DNA damage response (DDR) including cell cycle checkpoints[16]. If this persists the cell may ultimately die. To combat these challenges, eukaryotes have evolved both DNA repair as well DNA damage tolerance (DDT) mechanisms. One route of DDT is error-prone translesion synthesis (TLS). TLS allows DNA replication to continue past impediments and facilitates the filling of gaps which remain at the end of S-phase[17]. TLS utilizes specialized polymerases with active sites that can accommodate damaged or distorted DNA templates and which lack proofreading activity, hence increasing the risk of DNA mutagenesis[18,19]. In the germ cell compartment, ensuring that replication can proceed is of paramount importance to allow sufficient cellular expansion to guard against sterility. However, any increase in the mutagenicity of replication increases the risk of deleterious phenotypes and inherited disease in future generations.

Recent genome wide association studies (GWAS) have implicated multiple DDR pathways as determinants of infertility in humans, however little is known about the underlying mechanism or if this requirement extends to the TLS pathway[20–22]. In this study, we find that TLS plays a crucial role in the development of embryonic germ cells in both humans and mice. Our results show that factors involved in sequential stages of TLS are essential for PGC development. In the absence of TLS, PGCs are specified in normal numbers but fail to expand, resulting in a > 150-fold reduction. In contrast to the severe effect on the germline, somatic tissues of these mutants are unperturbed in their development with no discernible impact on homeostasis or survival. Consistent with a defect in TLS, mutant PGCs show reduced proliferation, accumulation at the G2/M phase of the cell cycle and increased markers of unresolved DNA damage. In addition, the loss of TLS prevents progression of the germ cell transcriptional program and results in failure of genome-wide DNA demethylation, an essential and conserved step in the development of mammalian embryonic germ cells. Our findings define a critical role of TLS specifically in germ cell development, safeguarding fertility and enabling successful PGC epigenetic reprogramming.

## Results

### REV1 is required for PGC development in human and mouse

GWAS studies focused on infertility have revealed genes important for human germline development[20–22]. Notably, factors that deal with DNA damage are frequent hits. In agreement with these findings, the DDR has been implicated in germ cell production and maintenance. Loss of several DDR factors results in infertility due to failure during meiosis, likely as a result of the inability to resolve meiotic DNA double strand breaks (DSBs) and recombination intermediates[23–29]. However, it has been found that DDR pathways and factors including Fanconi Anemia crosslink repair, base excision repair and homologous recombination amongst others are also required for the embryonic development of germ cells[21,30–38]. One of these factors, REV7, is involved in multiple DNA repair transactions including the repair of DNA DSBs, mitotic progression, and TLS[39–46]. Due to the requirement of various replication-coupled repair pathways in the embryonic germline, we set out to test if the TLS pathway is required for fertility. TLS is important for the completion of DNA replication, and we therefore hypothesized that it may play a role in ensuring successful genome duplication during highly proliferative stages of gametogenesis. As early embryonic germ cell development is particularly proliferative, we asked if TLS is required during primordial germ cell (PGC) development. We employed an in vitro model in which iPSCs are differentiated into human PGC-like cells (hPGCLCs) (Fig. 1a)[47]. To study the potential roles of TLS in human germ cell development we focused on REV1, a core component of the TLS pathway[48–52]. REV1 was disrupted in human BTAG iPSCs carrying both the BLIMP1-tdTomato and TFAP2C-EGFP PGCLC reporters (Supplementary Fig. 1a)[47]. Clones with successful disruption of REV1 were identified by PCR then validated by testing for

hypersensitivity to mitomycin C (MMC) (Supplementary Fig. 1b, c). We first assessed if the loss of REV1 affected iPSC function. As REV1 plays a role in DNA replication we assessed proliferation of mutant iPSCs and found no difference between $REV1^{-/-}$ and the parental line (Supplementary Fig. 1d). We also assessed the ability of $REV1^{-/-}$ iPSCs to differentiate into different lineages by measuring embryoid body formation[53]. We found that REV1 lines were able to differentiate into all three germ layers similarly to the parental wildtype line (Supplementary Fig. 1e). In contrast, we found that three independent $REV1^{-/-}$ lines had a significant reduction in the ability to induce hPGCLCs, consistent with a role in embryonic germ cell development (Fig. 1b, c and Supplementary Fig. 1f). While the frequency of hPGCLCs generated was significantly reduced in the absence of REV1, the few induced hPGCLCs expressed the canonical germ cell markers SOX17, TFAP2C and OCT4 (Fig. 1d). The magnified inset shows the nuclear localization of each factor (Fig. 1d).

Due to limitations of studying human gametogenesis in vivo, we asked if the requirement for TLS factors in PGC development was conserved to mice, thus facilitating mechanistic studies. REV1-deficient mice were crossed with wildtype mates to assess fertility. Consistent with our observations in human, we found that neither male nor female $Rev1^{-/-}$ mice gave rise to offspring despite evidence of copulation (Fig. 1e)[52]. Upon analysis of the gonads, we observed a striking reduction in testis mass but the mass of the ovaries was unaffected (Fig. 1f, g). Histological analysis of the testes revealed a majority (96.8%) of Sertoli-cell-only (SCO) seminiferous tubules and an absence of promyelocytic leukemia zinc finger (PLZF)+ cells, which marks undifferentiated spermatogonial stem cells (SSCs, Fig. 1h, i and Supplementary Fig. 1g)[54]. In females, we observed no follicles upon histological examination of $Rev1^{-/-}$ ovaries, suggesting a complete failure of oogenesis (Fig. 1h, j). Together, these data reveal a lack of gametes in both $Rev1^{-/-}$ male and female mice. The lack of visible PLZF+ cells in male gonads and meiotic cells in either sex argues that the defect is pre-meiotic which is consistent with the hPGCLC data suggesting a failure in germ cell development during embryonic stages. Therefore, we studied the development of embryonic germ cells using the GOF18-GFP PGC reporter in which GFP expression is driven by a fragment of the $Oct4$ promoter ($Oct4\Delta PE$) (Fig. 1k)[55,56]. We intercrossed $Rev1^{+/-}$ mice carrying the GOF18-GFP PGC reporter and harvested the embryonic gonads at E12.5, prior to sexually divergent germline development. Imaging of the gonads revealed a dramatic reduction in GOF18-GFP+ cells in $Rev1^{-/-}$ embryos compared to wildtype controls, consistent with the hPGCLC data (Fig. 1l). Quantification of PGCs (double positive for the marker SSEA-1 (stage specific embryonic antigen-1) and GFP; SSEA-1+GOF18-GFP+) by flow cytometry revealed a significant reduction in $Rev1^{-/-}$ embryos (Fig. 1m, n). Together, these data show that the TLS factor REV1 plays a key role in embryonic germ cell development in both humans and mice.

### The requirement for the C-terminal domain of REV1 and the Polζ subunit REV7 links TLS to PGC development

REV1 can act enzymatically through its deoxycytidyl transferase activity which can drive mutagenesis and is important for immunoglobulin diversification[48,50,51]. Alternatively, it can act as a protein-scaffold through its C-terminal protein interaction domain which is able to recruit TLS factors to sites of lesion bypass[57]. This region of REV1 plays a crucial role in maintaining cellular resistance to DNA damage that impedes replication indicating that the C-terminus is the key region of REV1 for TLS transactions. To determine if the catalytic or recruitment function of REV1 is required for PGC development, we generated a catalytically dead allele of $Rev1$, in which the catalytic residues D568 and E569 were mutated to alanine ($Rev1^{AA}$) (Fig. 2a and Supplementary Fig. 2a)[58]. We also generated mice with a mutant $Rev1$ allele in which the DNA encoding the final 100 amino acids of REV1 was deleted ($Rev1^{CT}$) (Fig. 2a and Supplementary Fig. 2b, c)[57]. To confirm the

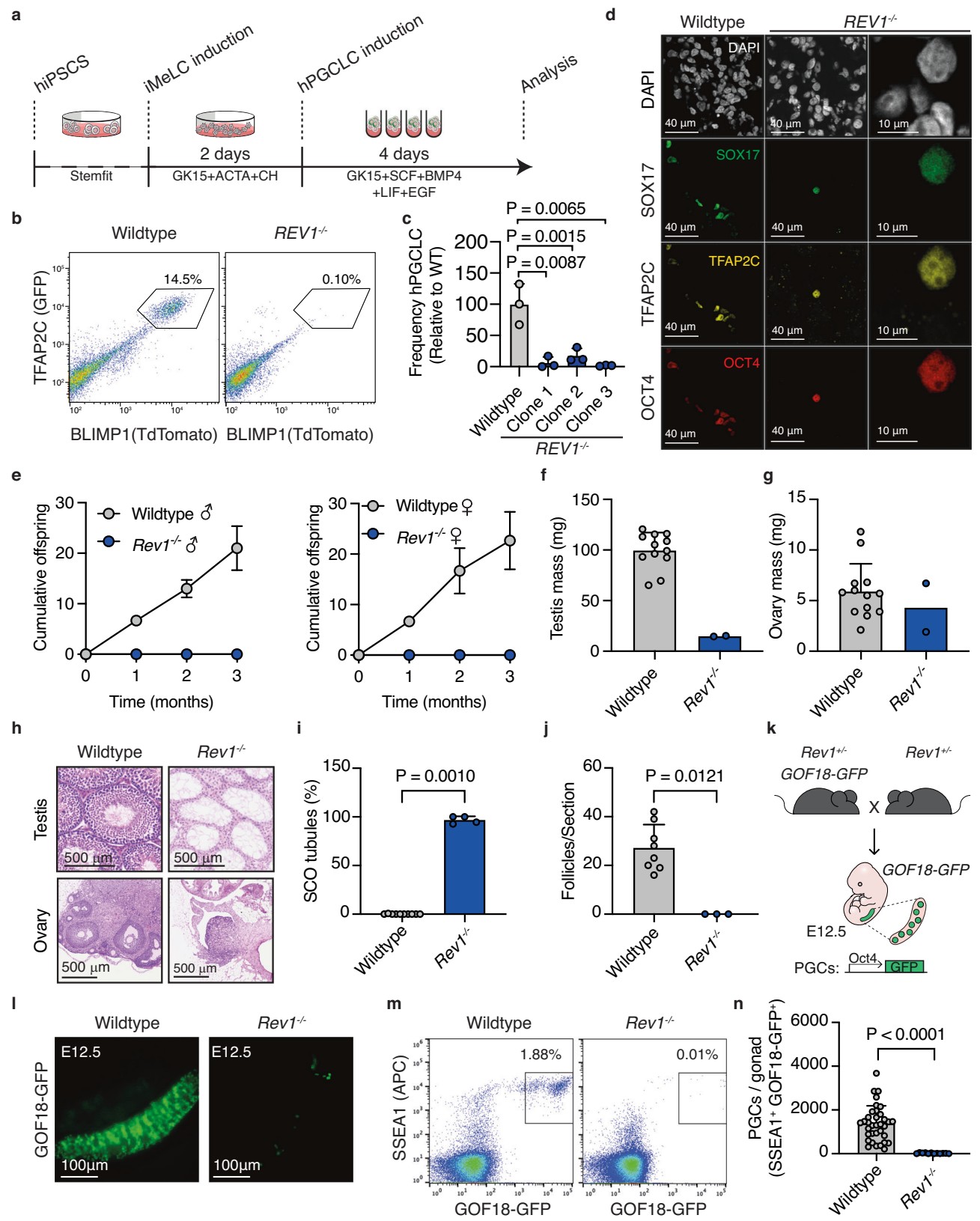

stability of the protein product of the *Rev1CT* allele, we expressed N-terminally FLAG-tagged full length or C-terminally truncated REV1 in cell lines and performed immunoblotting. This revealed comparable protein levels and a lack of degradation products (Supplementary Fig. 2d). Furthermore, immunofluorescence analysis of cell lines expressing GFP-tagged full length and C-terminally truncated

REV1 showed localization of both to the nucleus (Supplementary Fig. 2e). To further validate these *Rev1* mutant alleles, we derived cell lines from mice and measured mRNA expression and cellular sensitivity to DNA damaging agents. The mRNA expression of mutant alleles was comparable to that of *Rev1* in wildtype cells (Supplementary Fig. 2f). Moreover, the C-terminus but not catalytic activity of REV1 was

**Fig. 1 | REV1-deficiency leads to defects in hPGCLC induction and infertility in mice. a** Schematic of the hPGCLC differentiation protocol adapted from[47]. **b** Representative flow cytometry plots and (**c**) quantification of hPGCLC frequency at day 4 from three independent wildtype or *REV1*[-/-] clones differentiated three times. (Data represent mean and s.d. *P* values were calculated by a two-tailed Mann–Whitney *U*-test). **d** Representative images of wildtype and *REV1*[-/-] aggregates immunostained for SOX17, TFAP2C and OCT4. **e** Offspring when male (left) and female (right) wildtype or *Rev1*[-/-] mice were mated with wildtype mates of the opposite sex (*n* = 6 mice per genotype, 3 per sex. Data represent mean and s.d. *P* values were calculated by a two-tailed Mann–Whitney *U*-test). **f** Quantification of testis mass of 8–12-week-old wildtype and *Rev1*[-/-] mice (*n* = 12, 2, left to right. Data represent mean and s.d. *P* values were calculated by a two-tailed Mann–Whitney *U*-test). **g** Quantification of ovary mass from 8–12-week-old wildtype and *Rev1*[-/-] mice (*n* = 13, 2, left to right. Data represent mean and s.d. *P* values were calculated by a two-tailed Mann–Whitney *U*-test). **h** H&E-stained ovaries and testes seminiferous tubules from 8–12-week-old wildtype and *Rev1*[-/-] mice. **i** Quantification of SCO tubules per section of testis of 8–12-week-old wildtype and *Rev1*[-/-] mice (*n* = 10 and 4 animals, left to right. Data represent mean and s.d. *P* values were calculated by a two-tailed Mann–Whitney *U*-test). **j** Quantification of follicles per section of ovary from 8–12-week-old wildtype and *Rev1*[-/-] mice (*n* = 8 and 3 animals, left to right). Data represent mean and s.d. *P* values were calculated by a two-tailed Mann–Whitney *U*-test. **k** Schematic for generation of *Rev1*[-/-] embryos harboring the GOF18-GFP reporter. **l** GFP fluorescence images of gonads from wildtype and *Rev1*[-/-] E12.5 embryos. **m** Representative flow cytometry plots and (**n**) quantification of PGCs by flow cytometry from wildtype and *Rev1*[-/-] E12.5 embryos (*n* = 35 and 12, left to right. Data represent mean and s.d. *P* values were calculated by a two-tailed Mann–Whitney *U*-test).

required to overcome replication blocking DNA damage, in line with previous reports (Supplementary Fig. 2g)[57,59].

We next crossed the *Rev1*[AA] or *Rev1*[CT] alleles with *Rev1*[+/-] mice carrying the GOF18-GFP PGC reporter. At E12.5, we found that *Rev1*[-/AA] embryos had a reduction in the frequency of PGCs (median wildtype = 1383, median *Rev1*[-/AA] = 884, Fig. 2b, c). However, this was moderate when compared to *Rev1*[-/CT] or *Rev1*[-/-] embryos. Furthermore, *Rev1*[-/AA] PGCs generated mature gametes in the gonads of adult mice which were competent in giving rise to viable offspring (Fig. 2d–f and Supplementary Fig. 2h). In contrast, E12.5 *Rev1*[-/CT] embryos showed a significant reduction in the number of PGCs, with similar numbers to those in *Rev1*[-/-] embryos (median = 27 and 20, respectively, Fig. 2b, c). Together these data show that the catalytic activity of REV1 plays a moderate role in germ cell development whilst the C-terminus, which coordinates protein-protein interactions during TLS, is critical for PGC development.

To investigate the function of TLS in PGC development further, we investigated known REV1 interactors. First, we measured the expression of a subset of interactors (REV7, POLK and POLΘ) and found that like *Rev1*, each was more highly expressed in PGCs than in the surrounding somatic cells (Fig. 2g) (*29*, *37*). To determine if these factors are also required for PGC development, we utilized previously generated and validated genetic models to generate E12.5 embryos deficient in each factor expressing the GOF18-GFP PGC reporter[45,60,61]. Whilst *Polk*[-/-] and *Polq*[-/-] embryos had comparable numbers of PGCs to wildtype, REV7-deficient embryos had a significant reduction similar to that observed in *Rev1*[-/-] embryos (Fig. 2h, i). Consistent with this, the profound reduction in the number of PGCs in *Rev7*[-/-] embryos resulted in adult gonads devoid of germ cells (Fig. 2j–m and Supplementary Fig. 3a).

Together these data reveal that it is the C-terminal protein interaction domain of REV1 and its interactor REV7 that are required during PGC development. REV7/repro22 has previously been identified in an ENU screen as a factor needed for fertility and subsequent studies have shown REV7 to be important during PGC expansion[39–41]. REV7 is the non-catalytic subunit of Polζ, a B-family polymerase that extends the nascent strand following lesion bypass before handing back to replicative polymerases[62,63]. As only Polζ has such an activity it is considered essential for TLS transactions. However, REV7 also has non-TLS roles in DSB repair, cell cycle regulation and the shelterin complex[42–46]. For the polymerase activity of Polζ, REV7 must bind to REV3L which is not required for the other functions of REV7[64]. As REV3L is essential for embryonic development any potential role in PGC development cannot be tested[65–67]. However, we find that *Rev3l* and *Rev7* have similar patterns of expression in PGCs at E10.5 which mimics that of *Rev1* (Supplementary Fig. 3b). Crucially, Polζ is recruited to sites of lesion bypass for DNA synthesis through physical interactions between the REV7 subunit and REV1's C-terminal domain[46]. Hence, our discovery that both the C-terminus of REV1 and the REV7 subunit of Polζ are required in PGCs argues that the common function of both - in TLS - is needed during germ cell development.

## The post-translational modification of PCNA preserves PGC development

If TLS is critical for the development of PGCs, we hypothesized that PCNA ought to play an important role. PCNA is an essential component of the replisome that links DNA repair and DDT responses to replication[68]. Upon stalling of replication forks, lysine 164 (K164) of PCNA can be sumoylated, mono-, or poly-ubiquitinated to engage DDT pathways[69–71]. Monoubiquitination of PCNA at K164 is required for TLS, facilitating the recruitment of specialized polymerases to sites of replication blocking lesions, enabling their bypass[72–74]. As our results found a requirement for the TLS factors REV1 and REV7, we wanted to study if there is also a role for PCNA K164 modification during PGC development. We used CRISPR/Cas9 to generate mice in which K164 of PCNA is mutated to arginine (*Pcna*[K164R]) (Supplementary Fig. 4a). In this model, PCNA K164-modification dependent DDT, including TLS, is abolished[68]. We first set out to determine if our allele recapitulated features of previously described mice carrying the PCNA K164R mutation (*Pcna*[tm1Jcbs] MGI:3761720 and *Tg*[(Pcna*K164)1Mdsc] MGI:97503)[75,76]. As expected, *Pcna*[K164R/K164R] (herein referred to as *Pcna*[R/R]) mouse embryonic fibroblasts (MEFs) derived from our allele were hypersensitive to ultraviolet irradiation (Supplementary Fig. 4b)[75]. However, despite PCNA K164 being important for the cellular response to replication blocking DNA damage, *Pcna*[R/R] mice were born at the expected Mendelian ratios with adult mice having similar lifespan to wildtype littermates (Supplementary Fig. 4c, d). Consistent with previously published data, we found that the gonads of homozygous adult mice were smaller than wildtype and that both male and female *Pcna*[R/R] mice were sterile (Fig. 3a–d)[75]. Histological analysis revealed that the majority (98.2%) of the testes contained SCO seminiferous tubules and that the ovaries were devoid of follicles (Fig. 3e–h). Similar to REV1-deficiency, the failure in gametogenesis began during embryonic development (Fig. 3i, j). At E12.5, the numbers of PGCs observed in *Pcna*[R/R] embryos was similar to *Rev1*[-/-] with a > 150-fold reduction when compared to wildtype (Fig. 3k). To study if there were sex-specific differences in the embryonic germ cell defect of TLS-deficient embryos, we compared E12.5 male and female *Pcna*[R/R] embryos. We found comparable PGC numbers in both sexes, confirming a common defect (Supplementary Fig. 4e).

These findings led us to investigate the timing of the PGC defect in *Rev1*[-/-], *Rev7*[-/-] and *Pcna*[R/R] embryos more systematically. The PGCs in mouse embryos are specified between E6.0-6.5 then start to migrate and extensively proliferate from E8.5[1]. We first quantified the number of PGCs at E8.5 and found no significant difference in the number of PGCs between wildtype and the three TLS mutants (Fig. 3l). We confirmed this result using an alternative PGC-reporter, *Stella-GFP* (Supplementary Fig. 4f)[77]. Next, we quantified the number of PGCs by flow cytometry at each day of development between E9.5-12.5, replotting the E8.5 data on the same axis, and found that from E9.5 onwards *Rev1*[-/-] and *Pcna*[R/R] embryos had a significantly contracted PGC pool when compared to wildtype (Fig. 3m and Supplementary Fig. 4g–i). These

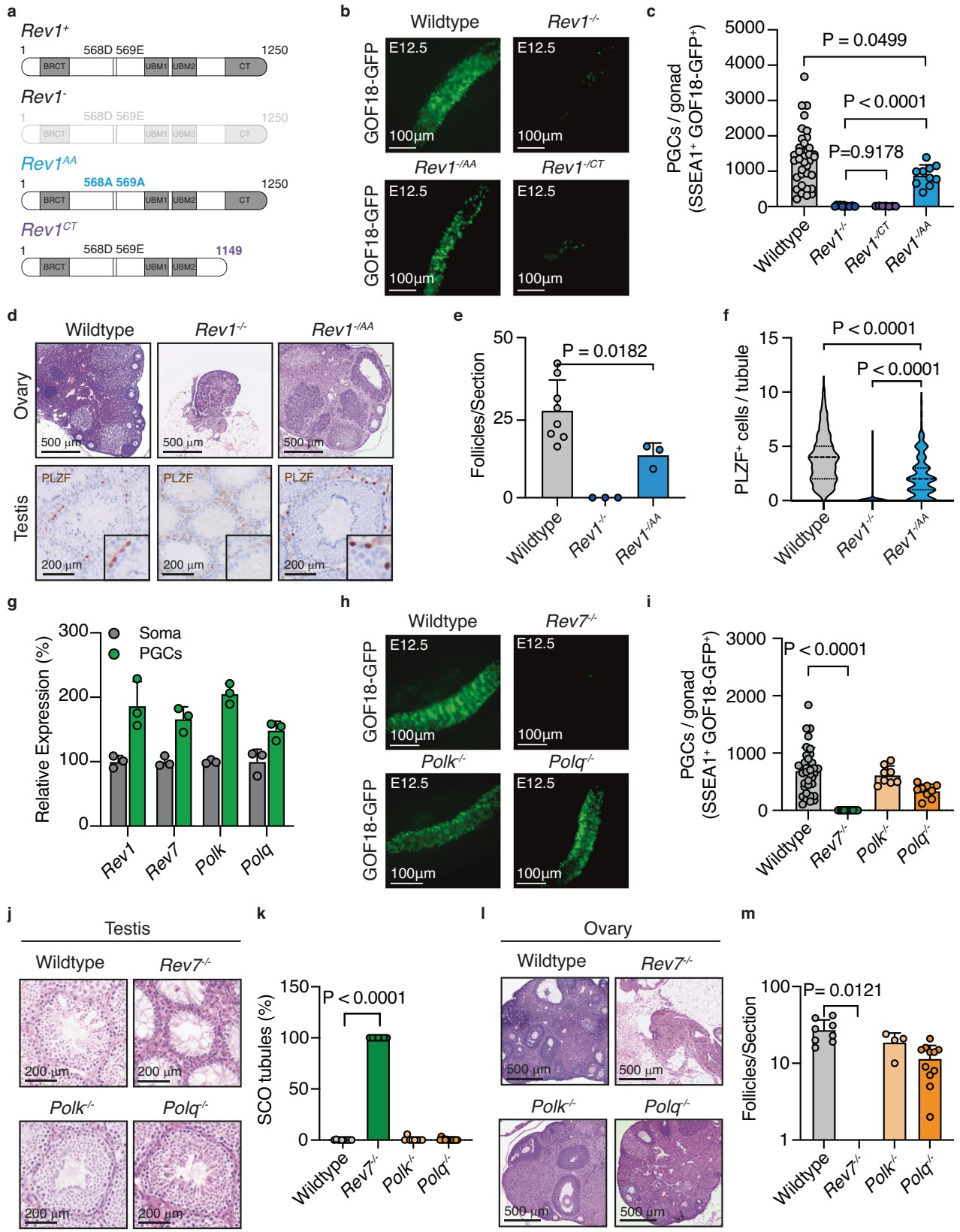

data reveal that in addition to similar magnitudes, the timing of PGC defect is comparable across the TLS mutants.

Having observed an essential requirement for TLS factors in mammalian germ cell development, we wondered if this was unique to the germline. In agreement with a previous report, we found a 2.6-fold reduction in the frequency of hematopoietic stem and progenitor cells (HSPCs) in adult $Pcna^{R/R}$ mice (Supplementary Fig. 5a, b)[78]. We went on to assess genome stability in the hematopoietic compartment by measuring the frequency of micronucleated normochromic erythrocytes (Mn-NCE). We found a significant increase in the frequency of Mn-NCE in $Pcna^{R/R}$ mice when compared to wildtype (Supplementary Fig. 5c). However, there was no change in the frequency of

**Fig. 2 | REV7 is required for PGC development in mice. a** Schematic of wildtype (*Rev1*[+]), null (*Rev1*), catalytically inactive REV1 (*Rev1*[AA]) and C-terminally truncated REV1 (*Rev1*[CT]) alleles. **b** GFP fluorescence images of gonads from wildtype, *Rev1*[-/-], *Rev1*[-/AA] and *Rev1*[-/CT] E12.5 embryos. **c** Quantification of PGCs by flow cytometry from wildtype, *Rev1*[-/-], *Rev1*[-/AA] and *Rev1*[-/CT] embryos at E12.5 (*n* = 35, 12, 10 and 7, left to right. Data represent mean and s.d. *P* values were calculated by a two-tailed Mann–Whitney *U*-test). **d** H&E-stained ovaries and PLZF-stained testes and quantification (**e**) of follicles per section of ovary (Data represent mean and s.d. *P* values were calculated by a two-tailed Mann–Whitney *U*-test). or (**f**) frequency of PLZF[+] cells per seminiferous tubule of 8–12-week-old wildtype, *Rev1*[-/-] and *Rev1*[-/AA] mice (the data shown represent the median and interquartile range; *n* = 150 tubules per genotype, 50 per genotype, *P* values were calculated by a two-tailed Mann–Whitney *U*-test). **g** Droplet digital PCR (ddPCR) gene expression analysis of *Rev1*, *Rev7*, *Polk* and *Polq* in FACS-purified PGCs and surrounding somatic cells (SSEA1[-]GOF18-GFP[-])

from E10.5 embryos (*n* = 3 independent embryos. Data represent mean and s.d. *P* values were calculated by a two-tailed Mann–Whitney *U*-test). **h** GFP fluorescence images of gonads from wildtype, *Rev7*[-/-], *Polk*[-/-] and *Polq*[-/-] E12.5 embryos. **i** Quantification of PGCs from wildtype, *Rev7*[-/-], *Polk*[-/-] and *Polq*[-/-] E12.5 embryos by flow cytometry (*n* = 35, 14, 8 and 9, left to right. Data represent mean and s.d. *P* values were calculated by a two-tailed Mann–Whitney *U*-test). **j** H&E-stained testis seminiferous tubules from 8–12-week-old wildtype and mutant mice. **k** Quantification of SCO tubules per section of testis of 8–12-week-old mice (*n* = 10, 7, 8 and 14, left to right. Data represent mean and s.d. *P* values were calculated by a two-tailed Mann–Whitney *U*-test). **l** H&E-stained ovaries and (**m**) quantification of follicles per section of ovary from 8–12-week-old mice (*n* = 8, 3, 4 and 11, left to right. Data represent mean and s.d. *P* values were calculated by a two-tailed Mann–Whitney *U*-test).

peripheral red blood cells or hemoglobin concentration (Supplementary Fig. 5d, e). As adult HSCs are largely quiescent we hypothesized that the defect may begin during embryonic development when HSPCs are highly proliferative[79]. Similar to adult mice, we found that E12.5 *Pcna*[R/R] embryos had a 1.73-fold reduction in the frequency of HSPCs, revealing that the blood stem cell defect begins during embryonic development (Supplementary Fig. 5f). Despite a numerical HSPC defect and evidence of genomic instability, we found that blood compartment homeostasis was remarkably intact with adult *Pcna*[R/R] mice showing comparable bone marrow cellularity to littermates, normal blood cell maturation, and sustained peripheral blood homeostasis (Supplementary Fig. 5g–l).

We next examined a panel of vital somatic tissues and found no gross histological abnormalities consistent with the normal longevity of *Pcna*[R/R] mice (Supplementary Fig. 6a). As PCNA is an essential component of the replication machinery, and its modification at K164 directly couples DDT to the replisome, we assessed the proportion of actively dividing cells in highly mitotic tissues. We found comparable numbers of Ki67[+] cells in the bone marrow, crypts of the ileum and hair follicle bulges of the skin of *Pcna*[R/R] mice compared to controls (Supplementary Fig. 6b–d). Furthermore, we also assessed the effect of PCNA K164R mutation on the function of the liver and kidney and found no difference compared to wildtype (Supplementary Fig. 6e–h). These data show that the K164R mutation of PCNA does not lead to a global reduction in proliferation nor to loss of tissue homeostasis or function. In contrast, histological analysis of the adult gonads revealed complete loss of homeostasis (Supplementary Fig. 7). Consistent with the histological analysis, we observe loss of tissue function in mutants leading to disruption of the hypothalamic-pituitary-gonadal axis. This led to systemic hormonal dysregulation with elevated levels of luteinising hormone (LH) and follicle stimulating hormone (FSH) driving reactive stromal hyperplasia and a significant increase in testicular interstitial cells and ovarian stroma. As animals age, persistent hypergonadotrophism leads to a substantial increase in ovarian mass with mutant ovaries of 12-month-old mice being 3 times larger than wildtype controls (Supplementary Fig. 7d). These features are consistent across TLS-deficient mice with similar defects seen in *Rev1*[-/-] and *Rev7*[-/-] adults (Supplementary Fig. 7).

Overall, whilst the PCNA K164R mutation results in increased sensitivity to exogenous DNA damage and a reduction in HSPCs, our findings reveal that the development and homeostasis of somatic tissues are largely unaffected. In contrast, the modification site of PCNA is essential for PGC development and fertility. Failure of embryonic germ cell development results in dysregulation of hypothalamic-pituitary-gonadal hormone regulation with inappropriate stromal proliferation in adults. The temporality and magnitude of *Rev1*[-/-], *Rev7*[-/-] and *Pcna*[R/R] germ cell defects and the molecular dissection of REV1 suggest a common origin of the defect - likely their shared role in TLS.

## Loss of genome stability and reduced proliferation of Pcna[R/R] and Rev1[-/-] PGCs

Next, we investigated the mechanisms of PGC failure in the absence of TLS factors. As TLS mitigates replication blocks, we asked if *Pcna*[R/R] or *Rev1*[-/-] PGCs accumulate unresolved DNA damage. We stained PGCs for the phosphorylation of histone variant H2A.X (γ-H2A.X), a marker of DNA DSBs[80]. A greater proportion of TLS-deficient PGCs had >10 γ-H2A.X foci compared to wildtype (Fig. 4a). Consistent with our previous data this finding was specific to PGCs as an increase was not observed in the surrounding somatic tissue (Supplementary Fig. 8a). To build on this, we stained for the presence of RPA foci, which binds to single stranded DNA generated during repair transactions or when DNA replication is perturbed and leads to under-replicated regions in the genome[81]. We stained for the RPA subunit RPA32 and quantified the frequency of PGCs harboring RPA foci as this is indicative of DNA damage[82,83]. Consistent with the results from the γ-H2A.X staining we observed a higher frequency of PGCs with nuclear RPA foci in *Pcna*[R/R] and *Rev1*[-/-] embryos (Fig. 4b). We went on to assess if the DDR was activated by measuring threonine 68 (Thr68) phosphorylation of CHK2, the critical checkpoint kinase[84]. We found that an increased frequency of *Pcna*[R/R] and *Rev1*[-/-] PGCs stained positive for pCHK2 when compared to wildtype (Fig. 4c). This shows that PGC development requires TLS and that in its absence cells accumulate damaged DNA.

Altered proliferation and apoptosis are frequent cellular responses to persistent DNA damage. The failure of the PGC pool to expand between E8.5-E12.5 in TLS-deficient embryos could be explained by either reduced PGC proliferation, increased PGC death, or a combination of both. We therefore first asked if TLS-deficiency leads to increased PGC death by staining E12.5 urogenital ridges for the apoptotic marker cleaved-caspase 3 (CC3)[85]. We observed significant 7.7-fold and 5.1-fold increases in the proportion of apoptotic (CC3[+]) PGCs in *Pcna*[R/R] and *Rev1*[-/-] embryos respectively (Fig. 4d). In contrast, there was no significant increase in the proportion of CC3[+] somatic cells in either mutant (Supplementary Fig. 8b). We next assessed in vivo PGC proliferation by injecting pregnant dams with a single dose of ethynyl-2′-deoxyuridine (EdU), which is incorporated into the DNA of replicating cells allowing the quantification of the proportion of cells in S-phase during the EdU pulse (Supplementary Fig. 8c)[86]. We found a significant reduction in the frequency of EdU[+] PGCs in both *Pcna*[R/R] and *Rev1*[-/-] embryos compared to wildtype (Fig. 4e). In the soma however, we observed a comparable number of EdU[+] somatic cells in mutant and wildtype embryos, suggesting that the reduced proliferation is restricted to the germ cell compartment (Supplementary Fig. 8d). To assess if reduced incorporation of EdU may be due to defects in cell cycle kinetics, we assessed the cell cycle properties of PGCs. First, we stained cells for Ser-10 phosphorylation of histone H3 (pH3) that occurs during cell division facilitating chromatin compaction, necessary for mitosis[87]. We found that a higher proportion of PGCs stained positive for pH3 in *Pcna*[R/R] and *Rev1*[-/-] genital ridges (Fig. 4f). We also

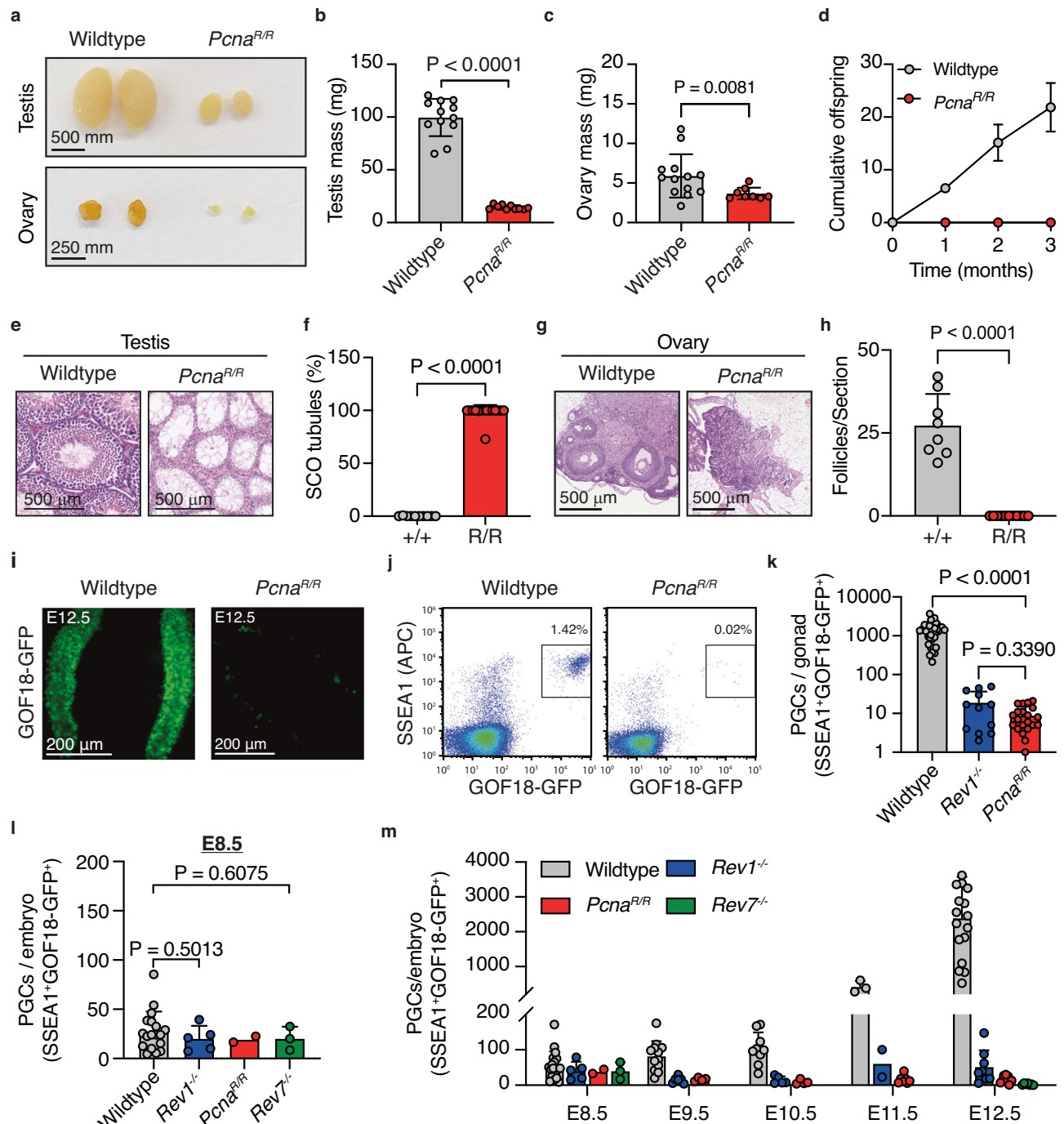

**Fig. 3 | Embryonic origin of sterility upon PCNA K164 mutation.**
**a** Representative images of testes and ovaries from wildtype and *Pcna^{R/R}* mice.- Quantification of (**b**) testicular or (**c**) ovarian mass from 8-12-week-old wildtype and *Pcna^{R/R}* mice (*n* = 12, 10, 13 and 8 left to right. Data represent mean and s.d. *P* values were calculated by a two-tailed Mann−Whitney *U*-test). **d** Cumulative number of offspring when wildtype or *Pcna^{R/R}* mice were mated with wildtype mates of the opposite sex (*n* = 6 mice per genotype, 3 per sex. Data represent mean and s.d. *P* values were calculated by a two-tailed Mann−Whitney *U*-test). **e** H&E-stained testis seminiferous tubules and (**f**) quantification of SCO tubules per section of testis of 8−12-week-old wildtype and *Pcna^{R/R}* (*n* = 10 and 15, left to right. Data represent mean and s.d. *P* values were calculated by a two-tailed Mann−Whitney *U*-test). **g** H&E-stained ovaries and (**h**) quantification of follicles per section of ovary from 8−12-week-old wildtype and *Pcna^{R/R}* mice (*n* = 8 and 11, left to right. Data represent mean

and s.d. *P* values were calculated by a two-tailed Mann−Whitney *U*-test). **i** GFP fluorescence images of gonads from wildtype and *Pcna^{R/R}* E12.5 embryos. **j** Representative flow cytometry plots and (**k**) quantification of PGCs from wildtype, *Rev1^{-/-}* and *Pcna^{R/R}* gonads at E12.5 (*n* = 35, 12 and 21, left to right. Data represent mean and s.d. *P* values were calculated by a two-tailed Mann−Whitney *U*-test). **l** Quantification of PGCs by flow cytometry from wildtype, *Rev1^{-/-}*, *Pcna^{R/R}* and *Rev7^{-/-}* embryos at E8.5 (*n* = 18, 5, 2 and 3, left to right. Data represent mean and s.d. *P* values were calculated by a two-tailed Mann−Whitney *U*-test). **m** Quantification of PGCs by flow cytometry from E8.5 to E12.5 in wildtype and mutant embryos (wildtype, *n* = 18, 12, 9, 3 and 17; *Rev1^{-/-}*, *n* = 5, 6, 5, 2 and 8; *Pcna^{R/R}*, *n* = 2, 4, 4, 7 and 9; *Rev7^{-/-}*, I = 3, 0, 0, 0 and 7, independent embryos, left to right. Data represent mean and s.d. *P* values were calculated by a two-tailed Mann−Whitney *U*-test).

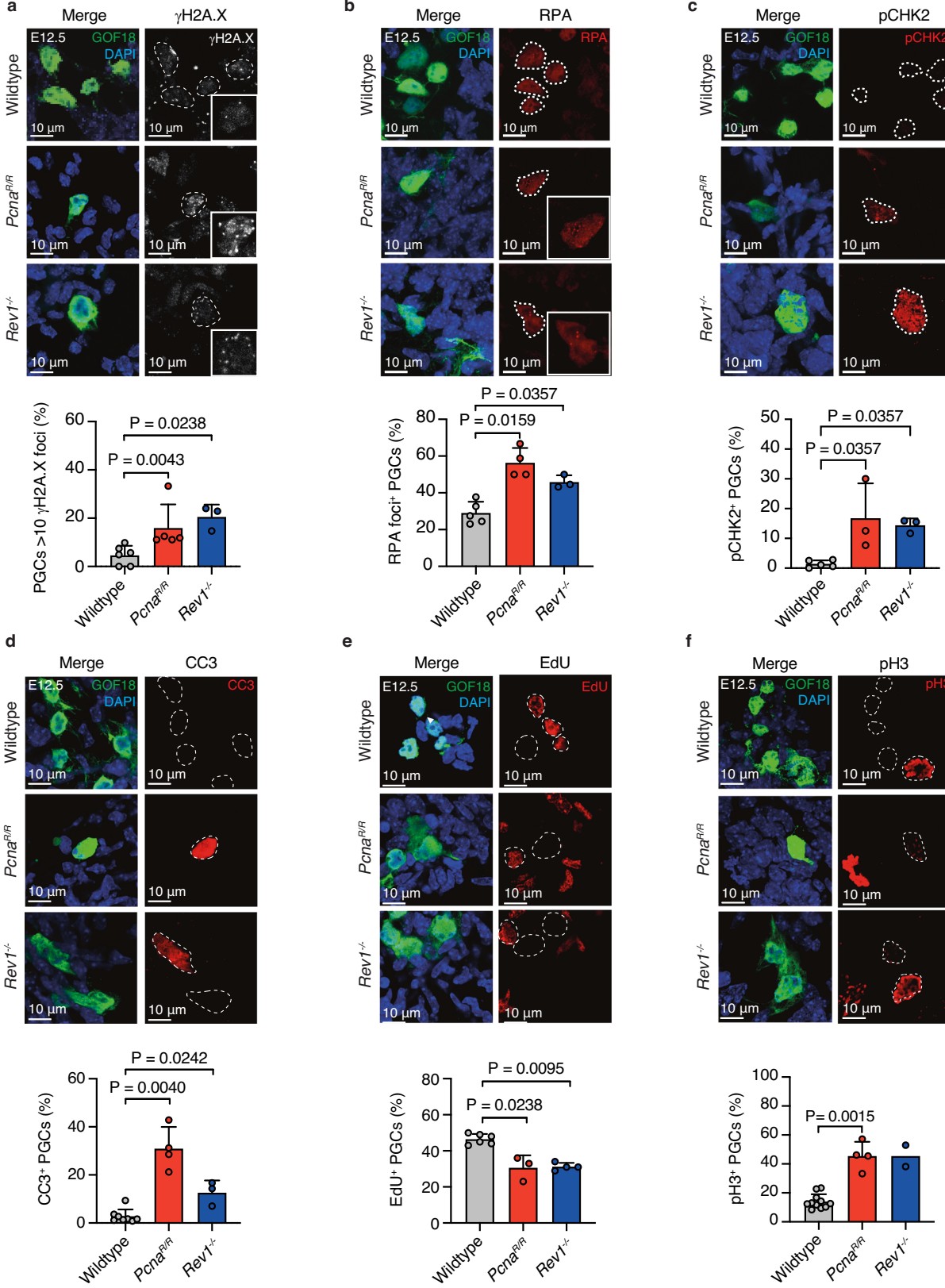

looked at the localization of cyclin B1 which is in the cytoplasm during G2, before becoming phosphorylated during mitosis, driving its relocalization to the nucleus[88,89]. At E12.5 15.7% of wildtype PGCs had nuclear cyclin B1 and were therefore in G2/M-phase (Supplementary Fig. 8e). This was significantly higher in the absence of TLS with a 2.6-fold increase in G2/M-phase PGCs in *Pcna^{R/R}* gonads (Supplementary

Fig. 8e). These data reveal that *Pcna^{R/R}* and *Rev1^{-/-}* PGCs accumulate markers of DNA damage, are less proliferative, accumulate at the G2/M phase of the cell cycle and are more likely to undergo programmed cell death. The combination of these cellular defects likely explains the profound reduction in the number of PGCs in the absence of TLS factors.

**Fig. 4 | Genome instability and cell cycle abnormalities in PcnaR/R and Rev1-/-
PGCs. a** Top: Representative images of γ-H2A.X foci in the nucleus of PGCs (GOF18-
GFP⁺) at E12.5. Bottom: Frequency of PGCs with >10 γ-H2A.X foci per nucleus ($n = 6$,
5 and 3, left to right. Data represent mean and s.d. $P$ values were calculated by a
two-tailed Mann–Whitney $U$-test). **b** Top: Representative images of RPA foci in the
nucleus of PGCs at E12.5. Bottom: Frequency of PGCs with RPA foci ($n = 5$, 4 and 3,
left to right. Data represent mean and s.d. $P$ values were calculated by a two-tailed
Mann–Whitney $U$-test). **c** Top: Representative images of E12.5 gonads stained for
phosphorylation of CHK2 kinase at residue Thr68 (pCHK2). Bottom: Frequency of
PGCs that stain positive for pCHK2 ($n = 5$, 3 and 3, left to right. Data represent
mean and s.d. $P$ values were calculated by a two-tailed Mann–Whitney $U$-test).
**d** Top: Representative images of E12.5 gonads stained for cleaved-Caspase-3 (CC3)
and GFP. Bottom: Frequency of PGCs that stain positive for CC3 ($n = 8$, 4 and 3, left
to right. Data represent mean and s.d. $P$ values were calculated by a two-tailed
Mann–Whitney $U$-test). **e** Top: Representative images of E12.5 gonads stained for
EdU and GFP. Bottom: Frequency of PGCs that stain positive for EdU ($n = 3$ per
genotype. Data represent mean and s.d. $P$ values were calculated by a two-tailed
Mann–Whitney $U$-test). **f** Top: Representative images of E12.5 gonads stained for
phosphorylated-histone-H3 (pH3) and GFP. Bottom: Frequency of PGCs that stain
positive for pH3 ($n = 11$, 4 and 2, left to right. Data represent mean and s.d. $P$ values
were calculated by a two-tailed Mann–Whitney $U$-test).

## Pcna^R/R and Rev1^-/- PGCs are developmentally blocked

Despite their scarcity, there are a few remaining PGCs in *Pcna^R/R* and
*Rev1^-/-* embryos at E12.5 (medians *Pcna^R/R* = 7 and *Rev1^-/-* = 15). However,
adult mice are completely sterile suggesting that the remaining PGCs
are not able to generate functional gametes. Successful PGC devel-
opment requires the activation of the germ cell transcriptional pro-
gram coupled to global epigenetic changes[90–92]. We therefore set out
to test if *Pcna^R/R* and *Rev1^-/-* PGCs underwent these critical transcrip-
tional and epigenetic processes. Initially, we performed gene expres-
sion analysis on E12.5 PGCs and found that both *Pcna^R/R* and *Rev1^-/-* PGCs
expressed the early markers of PGC development *Nanos3* and *Prdm1* at
comparable levels to wildtype (Fig. 5a). Conversely, expression of the
later-stage marker *Mvh* was dramatically reduced in both mutants
(Fig. 5a). We extended the gene expression analysis to additional germ
cell specific genes in *Pcna^R/R* E12.5 PGCs. Consistent with the gene
expression data presented above, genes normally expressed in the
later stages of PGC development (*Dazl, Mili, Mael, Sycp3, Mov10l1,
Hormad1* and *Brdt*) had reduced expression compared to wildtype
which was not true for early-stage genes (*Stella* and *Fragilis*) (Fig. 5b).
Therefore, E12.5 TLS-deficient PGCs transcriptionally resemble earlier
stages of development.

The expression of germ cell specific genes in PGCs is activated
during epigenetic reprogramming, specifically through DNA deme-
thylation of gene promoters[14,90–92]. The process of DNA demethylation
occurs across the whole genome and is unique to the PGC compart-
ment in the embryo. Alongside gene expression regulation, DNA
demethylation is critical for imprint erasure and X-chromosome
reactivation, processes needed for germ cell function. Our gene
expression data prompted us to assess DNA methylation of PGCs. We
performed whole genome bisulfite sequencing (WGBS) on wildtype
and *Pcna^R/R* FACS-purified E12.5 PGCs and compared these to wildtype
E6.5 epiblast cells, which are the origin of PGCs (Fig. 5c). Compared to
epiblast cells, the genome of wildtype E12.5 PGCs had extremely low
levels of DNA CpG methylation, reflecting demethylation during PGC
development. In contrast, in *Pcna^R/R* PGCs we found a near complete
retention of DNA CpG methylation (Fig. 5c). We mapped the methy-
lation levels across the genome and found the retention in *Pcna^R/R*
PGCs was genome-wide (Fig. 5d).

DNA methylation does not proceed uniformly, with different
genomic features reaching the lowest level of methylation at different
times[7,14,91,92]. We therefore asked if only a subset of genomic features
retained DNA methylation in TLS mutants or if the defect was truly
global. Analysis of gene bodies and promoters showed that these were
hypermethylated in *Pcna^R/R* PGCs compared to wildtype, suggesting a
failure in the early, global phase of demethylation (Fig. 5e, f). We also
found that repeat elements (LINE-1, SINE elements, and endogenous
retroviruses) retained methylation in *Pcna^R/R* PGCs (Fig. 5f and Sup-
plementary Fig. 9a). We validated this result by locus-specific, targeted
bisulfite DNA sequencing of LINE-1 as it makes up ~20% of the genome
and again found increased methylation in both *Pcna^R/R* and *Rev1^-/-* PGCs
(Fig. 5g and Supplementary Fig. 10a–d). We went on to analyze features
which undergo DNA demethylation in the later wave and for which
demethylation is both characteristic of germ cell development but also

required for function. We found that imprinted differentially methy-
lated regions (DMRs) were hypermethylated in E12.5 *Pcna^R/R* PGCs,
including both maternally and paternally methylated DMRs (Supple-
mentary Fig. 9b). Thus, this analysis reveals that the later wave of DNA
demethylation also fails in the absence of TLS.

CGIs (CpG islands) in the promoters of genes associated with
germ cell specific processes have been shown to retain DNA methy-
lation until E11.5 and be demethylated in the later wave of PGC DNA
demethylation[14]. These genes are involved in meiosis and gamete
generation and are only expressed in germ cells but silenced and
methylated in somatic cells. We found that these promoters were
hypermethylated in mutant PGCs (Supplementary Fig. 9c). We next
analyzed CpG methylation of genes required for germ cell production
whose transcriptional activation is concurrent with promoter deme-
thylation during the later stages of PGC development (germline
reprogramming responsive genes - GRR genes)[92]. CpG methylation
analysis of a panel of 45 GRRs showed that these were largely deme-
thylated in E12.5 wildtype PGCs (Fig. 5h). In *Pcna^R/R* E12.5 PGCs however,
these were hypermethylated. We confirmed this by performing locus-
specific BS-Seq of the promoters of two GRR genes *Mili* and *Dazl*
(Supplementary Fig. 10e, f). It is striking that we observed both
reduced expression of the GRR genes *Mvh, Dazl, Mili, Mael, Sycp3,
Mov10l1, Hormad1* and *Brdt* and also hypermethylation of these loci in
*Pcna^R/R* PGCs (Fig. 5b, i and Supplementary Fig. 9d). This retention of
DNA methylation in TLS mutants may explain the reduced activation
of GRRs

We compared the levels of DNA CpG methylation in mutant and
wildtype E12.5 PGCs to the E6.5 epiblast. The retention of methylation
across a panel of genomic features revealed that E12.5 *Pcna^R/R* PGCs
were more akin to the epiblast than to wildtype PGCs (Fig. 5e–g).
Together, these data show that loss of TLS prevents global DNA
demethylation in PGCs with retention occurring at early and late
demethylated regions. Furthermore, E12.5 mutant PGCs appear trap-
ped at an earlier stage of development with a methylome similar to
E6.5 epiblast cells.

Next, we focused on understanding the basis for methylation
retention in the absence of TLS and its effect on expression of germline
genes. First, we studied the machinery that normally adds methylation
marks to DNA by either ensuring methylation patterns are inherited
during cell division or by depositing new methylation marks. Following
DNA replication, the newly synthesized DNA strand is unmethylated
and DNMT1 is recruited to hemimethylated CpG dyads and transfers a
methyl group to the unmethylated cytosine whilst DNMT3A and
DNMT3B adds methylation marks to unmethylated DNA to deposit de
novo methylation. We measured the expression of *Dnmt1, Dnmt3a* and
*Dnmt3b* and found no difference between wildtype and *Pcna^R/R* E12.5
PGCs suggesting that overexpression of the methyltransferases is not
responsible for DNA hypermethylation in the absence of TLS (Fig. 5j
and Supplementary Fig. 9e). The enzyme TET1 has been shown to play
important DNA demethylation dependent and independent roles in
regulating the expression of genes critical for gamete formation.
Notably, TET1 has previously been shown necessary for the tran-
scriptional activation and maintenance of low levels of methylation at

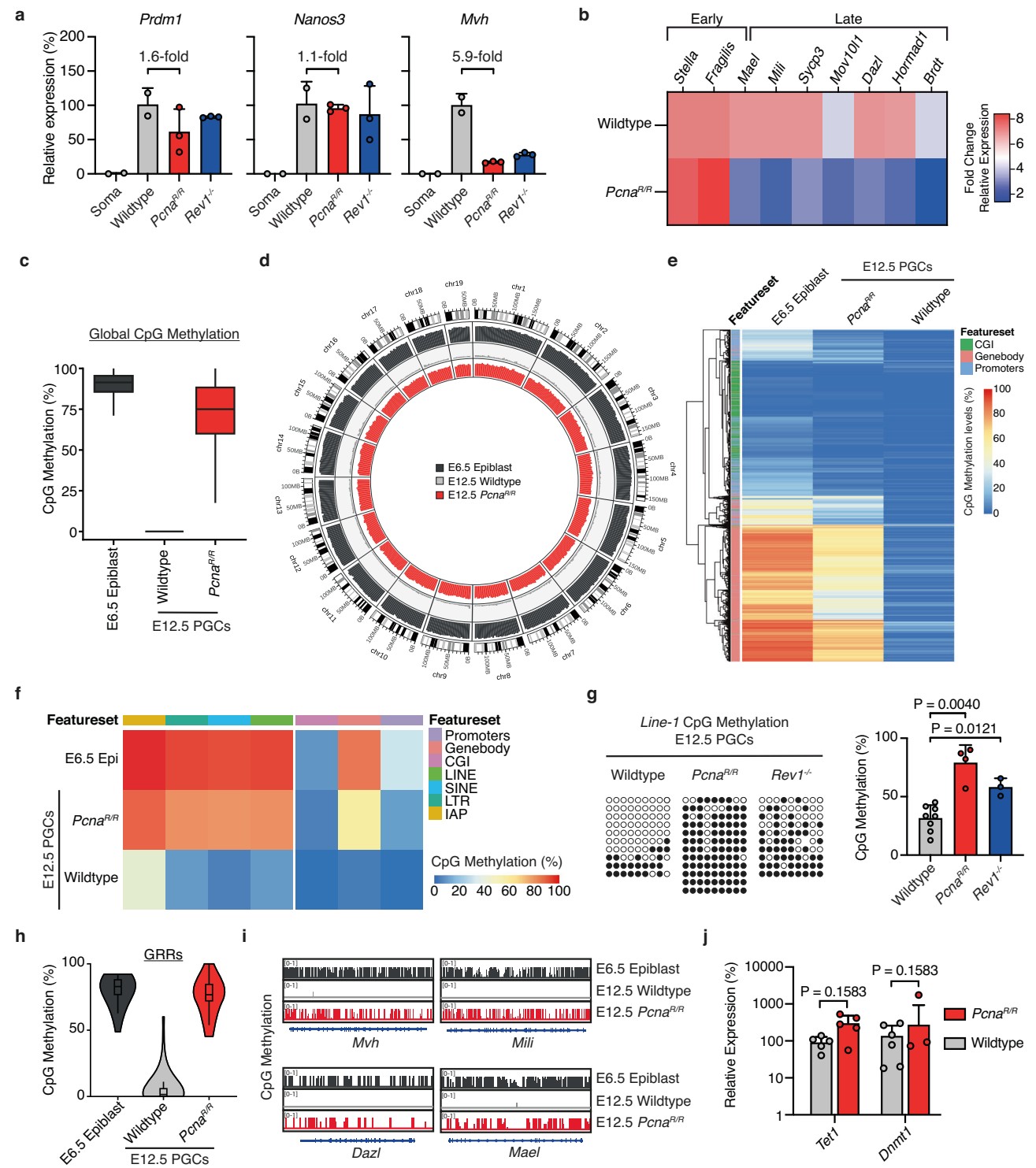

GRRs[92]. We quantified the expression of *Tet1* and its family member *Tet2* in wildtype and *Pcna^{R/R}* E12.5 PGCs and found no difference indicating that failure to activate the GRR genes is not mediated through *Tet1* expression (Fig. 5j and Supplementary Fig. 9f). We further validated this through TET1 immunostaining in E12.5 urogenital ridges which revealed the presence of protein in wildtype and *Pcna^{R/R}* E12.5 PGCs (Supplementary Fig. 9g).

Together, these data reveal that E12.5 TLS-deficient PGCs resemble earlier stages of germ cell development transcriptionally and in their methylome. As DNA methylation regulates gene expression it is plausible that the retention observed in TLS mutants

may contribute to the observed reduction in expression of germ cell genes. As well as failure of GRR demethylation, mutant PGCs also fail to erase imprinted DMR methylation, processes essential for normal germ cell function.

## Discussion

Declining birth rates and increasing infertility are drawing attention to environmental and genetic causes of human infertility. GWAS studies have identified the DDR as an important regulator of this, however mechanistic studies have for the most part been lacking. Our current work reveals a crucial role for DNA translesion synthesis during

**Fig. 5 | TLS preserves the PGC developmental programme. a** RT-qPCR expression analysis of PGCs from E12.5 embryos (*n* = 3 embryos per genotype. Data represent mean and s.d. *P* values were calculated by a two-tailed Mann–Whitney *U*-test). **b** RT-qPCR expression analysis of PGCs from E12.5 embryos. (Wildtype *n* = 6; *Pcna^R/R^* *n* = 7. Except for *Hormad1* and *Brdt*, wildtype *n* = 3; *Pcna^R/R^* *n* = 5). **c** Box plot of global DNA CpG methylation levels in E6.5 epiblast cells and E12.5 PGCs. The center displays the median, boxes the interquartile range and whiskers the minimum and maximum of CpG methylation distribution of the genome in 5Kbp genomic windows (*n* = 513027, 318836, 498368). **d** Circos-plot representation of DNA methylation levels in E6.5 epiblast and E12.5 wildtype PGCs and E12.5 *Pcna^R/R^* PGCs. CpG methylation was averaged in 5 Mbp genomic windows and the average DNA methylation is represented as a histogram track. **e** Heatmap showing methylation levels for E6.5 epiblast and E12.5 PGCs. **f** Unclustered methylation heatmap of CpG methylation in E6.5 epiblast and E12.5 PGCs. **g** Bisulfite sequencing and quantification of the *Line-1* element from E12.5 wildtype, *Pcna^R/R^* and *Rev1^-/-^* embryos (filled:methylated CpG, open:unmethylated CpG. Each point represents one embryo, data represent mean and s.d., *P* values were calculated by a two-tailed Mann–Whitney *U*-test). **h** Violin plots reflecting the DNA methylation levels of GRR gene bodies in E6.5 epiblast cells from wildtype embryos, E12.5 wildtype PGCs and E12.5 *Pcna^R/R^* PGCs. The center displays the median, boxes the interquartile range and whiskers the minimum and maximum of percentage methylation calculated over each gene body with each point representing an individual gene (*n* = 37, 37, 35 left to right). **i** CpG methylation across selected GRR genes in E6.5 epiblast and E12.5 PGCs. The plots represent the distribution of CpG methylation across genes segmented in 0.1 Kbp genomic windows. **j** RT-qPCR expression analysis of *Tet1* and *Dnmt1* in PGCs from E12.5 embryos. (For *Tet1*, *n* = 5 for both genotypes. *Dnmt1*, wildtype *n* = 6 and *Pcna^R/R^* *n* = 7. Data represent mean and s.d., *P* values were calculated by a two-tailed Mann–Whitney *U*-test).

germline development. Our findings demonstrate that REV1 is required for both human PGCLC and mouse PGC development, indicating a conserved embryonic cause of infertility.

Alongside REV1, we show that REV7 and the modification site of PCNA, K164, are essential for embryonic germ cell development. Though all three of these factors can act in multiple different processes, our results assert that it is their common role in TLS that is required[50]. REV1 has both catalytic and non-catalytic functions and we have found that the C-terminal protein interaction domain, critical for TLS, is required in PGCs. Like REV1, the monoubiquitination of PCNA at K164 serves as a TLS scaffold. REV7 interacts with REV3L to form the TLS extender polymerase called Polζ which has a specialized function in catalyzing DNA synthesis after lesion bypass during TLS. Importantly, Polζ is recruited to TLS sites through direct interactions between its REV7 subunit and REV1's C-terminal domain[50,57]. Our finding that both the REV1 C-terminus and REV7 are essential in PGCs supports a role for both in TLS. Our results did not identify an inserter TLS polymerase which may be explained by the well-characterized redundancy between the inserter polymerases[18,19]. We found that the three core TLS factors REV1, REV7 and PCNA K164 are required during embryonic stages of germ cell development and have identical timing and magnitude of PGC defects. In addition, the remaining PGCs in *Rev1^-/-^* and *Pcna^R/R^* embryos are phenotypically indistinguishable. This striking phenotypic overlap suggests that the cause of defect is the same in all three mutants with TLS being the only function common to all three factors. Whilst REV7 has described functions in DNA DSB repair and mitosis our data suggest that it is the role of REV7 in Polζ that is required for normal PGC development. Together, these data argue that TLS plays an essential role in fertility by preserving the development of PGCs.

DNA repair pathways such as Fanconi anemia (FA) DNA crosslink repair and DNA DSB repair, have previously been shown to be necessary for PGC development[30,32,36–38] (23, 25, 70–72). It is interesting that both genetic and biochemical studies have shown that the TLS factors REV1 and Polζ (REV7-REV3L) act in a common pathway with the FA repair proteins to maintain cellular resistance to DNA interstrand crosslinking agents[93–96]. Indeed we found that *Pcna^R/R^, Rev1^-/-^ and Rev7^-/-^* MEFs were mildly hypersensitive to DNA crosslinking agents, however this sensitivity was much less than in the absence of FANCA (Supplementary Fig. 11a). This does however leave the possibility that the loss of PGCs in FA-deficient and TLS-deficient mice shares a common cause. However, the magnitude and the temporality of the defects are different between the two classes of mutants with the loss of PGCs in TLS-deficient mice occurring 48 h before a loss of PGCs is observed in FA-mutants[30]. The magnitude of PGC loss in E12.5 TLS-deficient embryos is also two orders of magnitude greater than in the absence of the FA pathway[30]. Furthermore, there are substantial phenotypic differences between TLS-deficient and FA-deficient PGCs - the loss of FA repair factors does not alter the expression of GRR genes nor block DNA demethylation unlike the loss of TLS components[30]. This shows that whilst multiple DNA repair pathways are required for normal PGC development, they have distinct roles with different phenotypic outcomes.

The phenotypic features of the few remaining PGCs in TLS-deficient embryos explains the basis of germ cell loss and sterility. We found that mutant PGCs have a higher burden of DNA damage, apoptosis and cell cycle abnormalities as well as a failure in DNA demethylation and activation of the germ cell transcriptional program. Together these phenotypic features are likely to explain the basis of the failure of PGC development and infertility. Our results reveal that TLS is essential for the developmental processes that are needed for the production of mature germ cells[4,92].

The increase in DNA damage markers, whilst not previously shown in a physiological context, are consistent with known consequences of TLS deficiency upon genotoxin exposure in tissue culture systems. We observe an increase in the frequency of TLS-deficient PGCs with γ-H2A.X foci, RPA32 foci and CHK2 phosphorylation. Whilst none of these markers are entirely specific to the accumulation of DNA DSBs the combination of all three strongly suggests that in the absence of TLS PGCs accumulate DNA DSBs[84]. We also observed that an increased proportion of TLS-deficient PGCs are pH3-positive and have nuclear cyclin B1 localization, and therefore are at the G2/M phase of the cell cycle, with a reduced number staining positive for EdU following a 4 h pulse showing that fewer replicate their DNA and hence go through S-phase. These findings are consistent with a role for TLS either during S-phase (on-the-fly TLS) or in post-replicative repair. Whilst this data does not tell us when during the cell cycle DNA damage occurs it does suggest that it poses a block to mitosis following DNA synthesis unless it is resolved by TLS. It is interesting to note that embryonic stem cells have a less effective G1 checkpoint[97–100]. As this appears to also be the case in PGCs it would suggest that cells can enter into S-phase laden with DNA damage thus necessitating replication coupled repair such as TLS for resolution of damage and entry into mitosis[101]. Finally, we observed an increase in the proportion of TLS-deficient PGCs that undergo apoptosis which is a frequent cellular response to DNA damage. However, several studies have reported that loss of the DDR leads to a reduction in PGC number by altering the cell cycle rather than through apoptosis[102,103]. This difference may be due to the different repair pathways in these studies counteracting distinct classes of DNA damage that, when unrepaired, result in different cellular outcomes. However, apoptosis does play a role in the clearance of PGCs during unperturbed development or when epigenetic regulators, that normally suppress transposons, are lost[104–106]. Taken together, our data shows that in the absence of TLS PGCs accumulate markers of DNA damage, induce cell cycle arrest at the G2/M phase of the cell cycle and undergo apoptosis. It is likely that a combination of cell cycle arrest and apoptosis ultimately results in the numerical PGC defect. Whilst these findings are consistent with the described role of TLS

factors in dealing with DNA damage, the failure to undergo DNA demethylation or transcribe germ cell factors is entirely unexpected.

Whilst wildtype PGCs have very low levels of CpG methylation by E12.5, we found that TLS-deficient PGCs have genome-wide retention of DNA methylation. Indeed, we found that the methylation of TLS-deficient PGCs at E12.5 was similar to that seen in E6.5 epiblast cells. These data argue that the PGC precursors in the epiblast have undergone de novo methylation during early embryogenesis before later undergoing demethylation in PGC development. The observed cell cycle defects could explain why demethylation is prevented in the absence of TLS. Cell cycle arrested PGCs may be unable to respond to time-dependent cell extrinsic cues, such as signaling molecules that instruct PGCs to proceed in their development. Alternatively, the cell cycle perturbations may block a cell-intrinsic process, preventing activation of the germ cell program and epigenetic reprogramming. Interestingly, the first stage of global DNA demethylation in PGCs relies upon replication to dilute methylation marks[14,92]. Therefore, the cell cycle arrest in TLS-deficient PGCs may result in the genome-wide retention of methylation marks that we observe. The normal expression of TET1 in mutant PGCs argues against dysregulation of either its enzymatic or non-enzymatic functions as a contributing factor to the phenotype observed.

Whilst early stage PGC genes are induced by extrinsic signals and maintained by PGC-specific and pluripotency-associated transcription factors, DNA demethylation is important for the transcription of genes expressed in the later stages of PGC development. Therefore, the failure to transcribe GRR genes and retention of DNA methylation may not be independent of each other. Indeed, we find reduced mRNA expression and DNA hypermethylation of each GRR gene that was analyzed. It is therefore likely that the block in cell cycle progression in the absence of TLS prevents DNA demethylation, transcription of germ cell factors and therefore blocks PGC development.

The specificity of the phenotype which only affects germ cell homeostasis is intriguing. Despite loss of genome stability in the erythroid lineage and a reduction in blood stem cell frequency that we show begins in utero, TLS-deficient mice sustain peripheral blood homeostasis throughout life. Moreover, in vivo assessment of rapidly dividing tissues in $Pcna^{R/R}$ mice revealed no defects or overt phenotypes in somatic tissue homeostasis and consistent with this, mutant mice show normal longevity (Supplementary Figs. 5 and 6). In contrast, there is a complete failure to generate functional gametes. This striking dichotomy raises the question of why TLS is only required to maintain homeostasis in the germ cell compartment. We have demonstrated that TLS is required for processes that occur only in the germ cell lineage: engagement of germline genes and genome-wide DNA demethylation. In addition, we find that the canonical phenotypes associated with TLS deficiency and the DNA damage response are also restricted to the germ cell compartment: unresolved DNA damage, apoptosis, and cell cycle perturbation (Fig. 4 and Supplementary Fig. 8). This raises the possibilities that (i) the DNA damage checkpoints that activate apoptosis or cell cycle arrest are different in PGCs than in somatic cells, (ii) there are replication impediments or other features unique to PGCs that confer a dependency on TLS activity or (iii) that PGCs are more reliant upon TLS to resolve replication impediments than somatic cell types.

Future work will be required to determine which of these possibilities explains the selective requirement for TLS during PGC development. Recent work has shown that PGCs frequently experience transcription-replication conflicts and it is plausible that TLS factors are required to overcome such sources of genome instability[107,108]. Alternatively, the lack of a G1 cell cycle checkpoint in PGCs may render these cells uniquely dependent upon TLS to bypass damage during S-phase that in other cells would be repaired prior to the initiation of replication. Finally, it is tempting to speculate that the dependency of

PGC development on TLS may be directly connected to early epigenetic reprogramming events that are required for subsequent germline development.

# Methods

## Human iPSC culture
The hiPSC cell line (BTAG 585b1-868) was a kind gift from Prof. Mitinori Saitou (Institute for the Advanced Study of Human Biology, Kyoto, Japan). Cells were maintained in Stemfit Basic04 (Amsbio, SFB-504-CT) on Geltrex (Life Technologies, A1413302) coated dishes. Cells were passaged as single cells using Accutase (Millipore, SF006) and 10 µM of rock inhibitor (Y-27632, STEMCELL Technologies, 72304) was added for the first 24 h after seeding. Cells were maintained in hypoxic conditions (5% $CO_2$, 5% $O_2$) and without antibiotics.

## Induction of iMeLCs and hPGCLCs
Induction of iMeLCs and hPGCLCs was performed as previously published by Sasaki et al. (26). For iMeLC induction, hiPSCs were dissociated as single cells using a 1:1 mixture of TrypLE (Life Technologies, 12604013) and EDTA/PBS. $1.5 \times 10^5$ cells were seeded on a fibronectin coated well of a 12 well plate in GK15 medium (GMEM [Life Technologies] with 15% KSR, 0.1 mM NEAA, 2 mM L-glutamine, 1 mM sodium pyruvate, and 0.1 mM 2-mercaptoethanol) supplemented with: 50 ng/µl of ActivinA (R&D, 338-AC-010), 3 µM of CHIR99021 (Cambridge Bioscience, SM13-5) and 10 µM of Y-27632. For hPGCLCs, $3.0 \times 10^3$ iMeLCs were plated into ulta-low attachment U-bottom 96-well plate (PHCBI, MS-9096UZ) in GK15 supplemented with: 10 ng/ml LIF (Millipore, LIF1005), 200 ng/ml of BMP4 314-BP-010, 100 ng/ml of SCF (R&D Systems, 255-SC-050), 50 ng/ml of EGF (R&D Systems, 236-EG-200), and 10 µM of Y-27632.

## Generation of $Rev1^{-/-}$ BTAG hiPSC line
Two gRNAs were designed to target the exon 3 of REV1 (Supplementary Table 2) and subcloned into the PX458 plasmid (Addgene, 48138). $5.0-8.0 \times 10^5$ BTAG cells were electroporated with 3 µg of each gRNA using Amaxa 4D nucleofector (Lonza, V4XP-3012) and the CA-137 program. 48 h after the transfection, GFP positive cells were FACS sorted and plated in 96 well plate with one cell per well in Stemfit Basic04, 10 µM of Y-27632, penicillin and streptomycin (Life Technologies, 15070063). After 10−14 days single cell-derived colonies were picked and expanded. Clones were screened by PCR on extracted genomic DNA (NEB, T3010L) using the primer pair listed in Supplementary Table 2.

## Mitomycin C (MMC) treatment of BTAG hiPSC cells
Wildtype and $Rev1^{-/-}$ cells were passaged as single cells then seeded at 1:10 density in Stemfit Basic04 and Y-27632. 24 h after, medium was changed and Y-27632 was removed. The next day, 5 ng/ml MMC (Insight Biotechnology, sc-3514A) was added to the medium and 48−72 h later wells were assessed for surviving hiPSC colonies.

## Embryoid body formation assay
hiPSCs at 70−80% confluency were dissociated to a single cell suspension with Accutase. $2.4 \times 10^{-6}$ cells were seeded in an AggreWell™400 (Stemcell technologies) plate in embryoid body (EB) formation medium (Stemcell technologies, 05893) supplemented with Y-27632 rock inhibitor and incubated for two days at 37 °C with 5% $CO_2$. Newly formed EBs were collected from the wells and centrifuged at 160 x g for 3 min, resuspended in EB medium: DMEM/F-12 with Glutamax (ThermoFisher, 10565-018), 20% KSR (ThermoFisher, 10828010), 1x NEAA (ThermoFisher, 11140-050) and 2-Mercaptoethanol (ThermoFisher, 21985-023) then plated in 10 cm dishes and maintained at 37 °C with 5% $CO_2$. Medium was changed every other day for 21 days. EBs were collected, washed, trypsinised and used for RT-qPCR analysis.

## Embryoid body gene expression analysis

Total RNA was extracted from hiPSCs and their counterpart day 21 EBs using the RNeasy kit (Qiagen) according to the manufacturer's instructions. First-strand complementary DNA was synthesized using PrimeScript RT Reagent Kit (Takara) according to the manufacturer's instructions. PCR amplification was performed using the SensiMix SYBR No-ROX Kit (Bioline) on a CFX96 machine (Biorad). The primers used for this experiment are listed in Supplementary Table 3. Mean threshold values were determined using standard comparative $C_T$ methods. All expression levels were normalized to the hiPSC line used to generate the EBs.

## Proliferation assays

hiPSCs were plated into flat-bottomed 96-well plates (ThermoFisher, 167008) at a density of 1000 cells per well in Stemfit supplemented with Y-27632 rock inhibitor. Every day for 5 days the absorbance of the different samples was assessed. Briefly 20 µl of MTS reagent (Promega, G3582) was added to each well and the plate was incubated at 37 °C for two hours. Following incubation, absorbance was measured at 490 nm using a plate reader.

## hPGCLC immunostaining

Day 4 hPGCLC aggregates were fixed in 4% PFA for 20 min at room temperature. After PBS washes and incubation in 30% sucrose, aggregates were embedded in OCT (VWR, 361603E) and sectioned at a thickness of 10 µm. Samples were permeabilized for 20 min in PBS/0.2% Triton X100 (PBST) and blocked for 1 h in blocking solution (Santa Cruz, sc-516214). Samples were then incubated with the following primary antibodies diluted in blocking buffer at room temperature for 2 h: anti-OCT4 (1:200, catalog no. ab181557; Abcam); anti-TFAP2C (1:100, catalog no. sc-12762; Santa Cruz); anti-SOX17 (1:200, catalog no. AF1924; R&D). After PBS washes, samples were incubated with the following secondary antibodies diluted 1:1000 in PBS: Donkey anti-rabbit IgG (Alexa Fluor 594, catalog no. A21207; Invitrogen); Donkey Anti-Goat IgG H&L (Alexa Fluor 488, catalog no. ab150129; Abcam); Donkey anti-Mouse IgG (H&L) (Alexa fluor 594, catalog no. R37115; Life Technologies). Sections were then washed with PBS and incubated for 10 min with DAPI. A coverslip was then placed on the slides in Vectashield vibrance (Vector Labs, H-1800-10). Sections were imaged using the SP5 inverted confocal microscope (Leica).

## Flow cytometry analysis of hPGCLC aggregates

Day 4 aggregates were collected and dissociated with 0.25% trypsin-EDTA (Life Technologies, 25200056) for 10 min at 37 °C under agitation. Cells were washed with PBS containing fetal bovine serum (FBS) and 0.1% bovine serum albumin (BSA) before being subjected to centrifugation. Dissociated cells were then resuspended in FACS buffer (PBS, 0.1% BSA), filtered by a cell strainer (BD Biosciences) and analysed or sorted by FACS (Aria III, BD Biosciences) based on the eGFP and tdTomato reporters. Analysis of hPGCLCs differentiation efficiency was performed on one wild type clone and three REV1 KO clones in at least three independent experiments. FlowJo version 10 was used for analysis.

## Mice

All animal experiments performed in this study were approved by the Medical Research Council's Laboratory of Molecular Biology animal welfare and ethical review body and conform to the UK Home Office Animal (Scientific Procedures) Act 1986 (License no PP6752216). All mice were maintained under specific pathogen-free conditions in independently ventilated cages (GM500; Techniplast) on Lignocel FS-14 spruce bedding (IPS) and provided with environmental enrichment (fun tunnel, chew stick and Enviro-Dri nesting material (LBS)) at 19−23 °C. Mice fed Dietex CRM pellets (SpecialDietService) ad libitum with light from 7:00 a.m. to 7:00 p.m. No mice used in this study were

wild and no field-collected samples were used. All mice were maintained on an isogenic C57BL/6J background. Sex-specific analysis was performed for analysis of adult reproductive tissues as detailed in the text. Primordial germ cell analyses were performed before sexually divergent development of the germline and hence data from both sexes was combined. The lack of a sex-specific phenotype was confirmed through disaggregation and separate analysis of E12.5 PGC numbers for $Pcna^{R/R}$ mutants. Embryos were examined at various developmental stages from E8.5 to E13.5 as indicated in the text. Female mice used in timed-mating experiments were aged between 6-25 weeks. The investigators were blinded to the genotypes of all mice throughout the study and data were acquired by relying entirely on identification numbers. The *Stella-GFP* (*Tg(Dppa3/EGFP)6-25Masu*) allele (MGI ID: 5519126) was a kind gift from Azim Surani[77]. B6;129P2-*Polk^{tm1.1Rsky}/J* allele (MGI ID: 2445458) and the B6.Cg-*Polq^{tm1Jcs}/J* allele (MGI ID: 2155399) have been described previously[60,61]. *GOF18-GFP*(*Tg(Pou5f1-EGFP)2Mnn*) (MGI ID: 3057158) JAX (stock ID: 004654) mice were purchased from The Jackson Laboratory[55,56]. We imported the previously described C57BL/6NTac-Mad2l2<tm1a(EUCOMM) Wtsi > /WtsiCnbc strain (EM: 05374), frozen sperm from The European Mouse Mutant Archive and used to derive live mice[45]. These mice were maintained on a C57BL/6J background.

## Generation of *Pcna^{K164R}*, *Rev1^{AA}* and *Rev1^{CT}* mutant mice

Mice carrying the *Pcna^{K164R}* and *Rev1^{CT}* alleles were generated by Alt-R CRISPR-Cas9 (IDT) mediated genome editing in zygotes on a C57BL/6J background. tracrRNA and crRNA were diluted to a final concentration of 1 µg/ul in injection buffer (10 mM Tris HCl pH 7.5, 0.1 mM EDTA). 5 µg of crRNA and 10 µg tracrRNA were mixed and annealed by heating to 95 °C for 5 min and then ramped to 25 °C at a rate of 5 °C per min. Alt-R SpCas9-3NLS was diluted to a concentration of 200 ng/µl in injection buffer. The RNP was assembled by diluting Alt-R SpCas9-3NLS and the annealed crRNA:tracrRNA at a final concentration of 20 ng/µl and incubated at room temperature for 15 min. The ssODN was then added to the RNP complex at a final concentration of 20 ng/µl and injected into zygotes. The zygotes were surgically transplanted into pseudopregnant CD1 females. Progeny were screened for correct gene targeting and the targeted allele sequenced. *Rev1^{AA}* mice were generated by gene targeting in mouse embryonic stem cells. Stella-GFP Bac9 ESCs were transfected with an ssODN and px451 plasmid containing a guide targeting exon 10 of *Rev1*. Clones were screened by PCR followed by AciI restriction digest. Two positive ESC clones were injected into C57BL6/J:TYR blastocysts. Mice with high levels of chimerism were back crossed and the progeny screened by PCR.

## Isolation of mouse embryonic fibroblasts

Timed matings were performed between heterozygous mice. Pregnant females were culled at E12.5 to harvest embryos. Embryos were incubated in pre-warmed trypsin solution (2.5 µg.mL⁻¹ trypsin (Gibco), 25 mM Tris, 120 mM NaCl, 25 mM KCl, 25 mM KH₂PO₄, 25 mM glucose, 25 mM EDTA, pH 7.6) for 10 min and disaggregated by gentle pipetting. Primary mouse embryonic fibroblast (MEF) cultures were established following standard methods and immortalized using the SV40 large T antigen as described previously. Briefly, Platinum-E retroviral packaging cells (Cell Biolabs) were transfected with pBABE-SV40-Puro and the culture media containing the virus was harvested 48 h later and passed through a 0.22 µm filter. The filtered retrovirus was mixed 1:1 with complete MEF media supplemented with 1 µg.mL⁻¹ hexadimethrine bromide (Polybrene, Millipore). The infective medium was subsequently added to primary MEF cultures and transformed clones were selected for 14 days using 1 µg.mL⁻¹ puromycin.

## Measuring sensitivity of MEFs to DNA damaging agents

Sensitivity to DNA-damaging agents was determined by seeding 1000 transformed MEFs per well of a 96-well flat-bottom plate and exposing

to ultraviolet irradiation or mitomycin C (MMC). After 7 days of culture the MTS cell viability reagent (CellTiter 96® Aqueous One Solution Cell Proliferation Assay, Promega) was added and plates incubated for 4 h at 37 °C; absorbance was then measured at 492 nm.

## Histological analysis

Tissues were fixed in 10% neutral-buffered formalin for 24–36 h then transferred to 70% ethanol. Fixed samples were dehydrated and embedded in in paraffin and 4 μm sections cut. Sections were deparaffinised, re-hydrated and stained with haematoxylin and eosin (H&E) following standard methods. Images were captured with an Eclipse Ti2-E (Nikon) microscope and tissue architecture was scored blindly.

## Immunohistochemistry

Formalin-fixed, paraffin-embedded samples were sectioned at 4 μm, deparaffinised and rehydrated following standard methods. Slides were boiled in antigen retrieval buffer (10 mM sodium citrate, pH 6) for 10 min and allowed to cool to room temperature before being washed three times in water for 5 min and then once in TBS, 0.1% w/v Tween-20 for 5 min. A hydrophobic ring was drawn around the tissue sections and samples incubated with blocking buffer (TBS, 0.1% w/v Tween-20, 5% v/v goat serum) for 1 h at room temperature. Samples were incubated with the following primary antibodies diluted in blocking buffer at 4 °C overnight: anti-PLZF (1:200, sc-28319) and anti-Ki67 (1:200, ab16667). Slides were washed three times with TBS, 0.1% w/v Tween-20 for 5 min and incubated with the following secondary antibodies for 1 hour at room temperature: swine anti-rabbit (1:200, catalog no. P0339, Dako) or anti-mouse horseradish peroxidase (HRP)-conjugated immunoglobulins (1:200, catalog no. P0339, Dako). For HRP-based immunohistochemistry, slides were incubated for 3–10 min with SignalStain diaminobenzidine substrate kit (catalogue no. 8059 P; Cell Signaling Technology) and then washed once in water for 5 min. Slides were dehydrated in an ethanol gradient following standard methods and finally in xylene before being mounted with DPX neutral mounting medium (catalog no. 317616, Sigma-Aldrich) and coverslips place on top of the slides. Images were captured with an Eclipse Ti2-E (Nikon) microscope. The frequency of PLZF+ or Ki67+ cells were scored blindly.

## Assessing fertility of mice

Mice were paired with wildtype C57BL/6J mice of the opposite sex. Female mice were monitored daily for the presence of copulation plugs and the number of offspring born over three successive months was recorded (only data from breeding pairs where at least 3 copulation plugs were observed was included in the analysis). Investigators performing the copulation plug checks were blinded to the genotypes of the mice.

## Timed matings for embryo isolation

Timed matings were performed overnight and female mice were assessed for the presence of copulation plugs the following day and separated from males. Halfway through the light cycle on the day a copulation plug was observed was designated E0.5. Pregnant mice were culled at noon of the appropriate day during gestation (E8.5–13.5) and the embryos harvested. Samples were processed immediately for further analysis with a small tissue biopsy taken for genotyping.

## Immunofluorescence

Timed matings were performed as described above and E12.5 embryos harvested. The fetal gonads were dissected out and fixed in PBS, 4% w/v paraformaldehyde for 30 min at 4 °C. Fixed samples were washed three times in PBS, 1% w/v Triton X-100 for 5 min at room temperature for permeabilization then pressed onto glass slides. A large hydrophobic ring was drawn around each sample and then incubated in blocking buffer (PBS, 1% w/v Triton X-100, 1% w/v BSA) for 1 h at room temperature. Samples were then incubated with the following primary

antibodies at 4 °C overnight: anti-GFP (1:500, catalog no. GF090Rl; Nacalai); anti-GFP (1:2000, catalog no. ab13970); anti-phospho-Histone H2A.X (Ser139) (1:1000, catalog no. 05-636; Millipore); anti-cleaved caspase 3 (1:400, catalog no. 9661; Cell Signaling Technology); anti-MVH (ab27591); anti-RPA (1:100, catalog no, 2208; Cell Signaling Technology); anti-cyclin B1 (1:100, catalog no. 4138; Cell Signaling Technology); anti-phospho-Chk2 (Thr68) (1:100, catalog no, 2661; Cell Signaling Technology); anti-TET1 (1:100, ab272901) and anti-phospho-histone 3 (1:200, catalog no, 9701; Cell Signaling Technology). Samples were then washed three times in PBS, 1% w/v Triton X-100 for 5 min and then incubated for 1 h at room temperature with the following secondary antibodies: goat anti-rat Alexa Fluor 488 (1:1000, catalog no. A11006, Thermo Fisher Scientific), goat anti-chicken Alexa Fluor 488 (1:1000, catalog no. A11039, Thermo Fisher Scientific), goat anti-mouse Alexa Fluor 594 (1:1000, catalog no. A11032, Thermo Fisher Scientific), goat anti-rabbit Alexa Fluor 594 (1:1000, catalog no. A21429; Thermo Fisher Scientific), goat anti-rat Alexa Fluor 594 (1:1000, catalog no. A21209; Thermo Fisher Scientific). Slides were then washed three times in PBS, 1% w/v Triton X-100 for 5 min and stained with 0.5 μg.mL⁻¹ DAPI diluted in PBS for 10 min. Slides were washed once then mounted with ProLong Gold Antifade mounting media (catalog no. P36934; Thermo Fisher Scientific). Coverslips were placed on top of the slides and the slides were allowed to cure for 48 hours. Images were captured with an LSM780 confocal microscope (Zeiss) and specimens were scored blindly.

## Immunofluorescence on cultured cells

To analyze the localization and stability of C-terminally truncated REV1, the sequences for full length and truncated REV1 were cloned into pEGFP-C1. HEK293 cells were seeded on no. 1.5 coverslips (catalog no. 631-0150, VWR) and 24 h later transfected with either plasmid. Subsequently, cells were washed twice for 5 min with PBS then fixed with 4% paraformaldehyde (catalog no. 43368, Alfa Aesar) for 20 min and washed with PBS twice for 5 min. Cells were then permeabilised for 10 min with PBS containing 0.1% Triton X-100 and washed with PBS twice for 5 min. Cells were then blocked with PBS-S/0.1% Tween 20 (PBS-S-T) containing 5% BSA for 30 min. Cells were then incubated overnight at 4 °C with the following primary antibody diluted in blocking buffer: anti-GFP (1:500, catalog no. GF090Rl; Nacalai). Cells were then washed with PBS and incubated for 1 h at 37 °C with the following secondary antibody diluted in blocking buffer: goat anti-rat Alexa Fluor 594 (1:1000, catalog no. A21209; Thermo Fisher Scientific). After washes with PBS, coverslips were incubated for 10 min with 2 μg/ml DAPI. After washes in PBS, coverslips were mounted on glass slides using Prolong Gold Antifade Mountant (catalog no. P36934, Thermo-Fisher Scientific). Images were captured with a LSM780 confocal microscope (Zeiss).

## Immunoblotting

To demonstrate the stability of C-terminally truncated REV1, the sequences for full length and truncated REV1 were cloned into pExpress with a 2x FLAG tag at the N-termini. HEK293 cells were transfected with either plasmid and were harvested for protein extraction 48 h later. Protein samples were supplemented with LDS buffer (catalog no. NP0007, Themo Fisher Scientific) and 5% β-mercaptoethanol final, boiled for 5 min at 95 °C and resolved by polyacrylamide gel electrophoresis on NuPAGE 4–12%, Bis−Tris, Mini Protein gels (catalog no. NP0321BOX, ThermoFisher Scientific) in MOPS-SDS buffer (50 mM MOPS, 50 mM Tris base, 3.47 mM SDS, 1 mM EDTA). Separated proteins were transferred onto 0.2 μm nitrocellulose membranes (catalog no 10600015, GE Healthcare) in Tris-glycine (25 mM Tris, 192 mM glycine, ph 8.3) buffer with 20% ethanol. Transfer was set at 35 V for 90 min in a Xcell II Blot module (catalog no. EI9051, ThermoFisher Scientific). After transfer, membranes were incubated for 1 h in blocking buffer (Tris-buffered saline, 0.1% Tween 20, 5% non-fat dry

milk). Membranes were then incubated with following primary antibodies diluted in blocking buffer and incubated at 4 °C overnight with gentle agitation: anti-FLAG (1:200, M2 clone, catalog no. F1804, Sigma-Aldrich) and anti-β-actin (1:1000, catalog no. ab8227; Abcam). The following secondary antibodies were diluted in blocking buffer and incubated for 1 h at room temperature with gentle agitation: swine anti-rabbit Ig HRP-conjugated (1:3000, catalog no. P0399, Dako) and goat anti-mouse Ig HRP-conjugated (1:5000, catalog no. P0447, Dako).

### Preparation of PGCs and their quantification and isolation by flow cytometry

For PGC quantification, the entire embryo (E8.5-10.5) or the developing urogenital ridges (E11.5-13.5) were isolated from embryos and placed into 500 µL or 150 µL pre-warmed trypsin solution (2.5 µg.mL$^{-1}$ trypsin (Gibco), 25 mM Tris, 120 mM NaCl, 25 mM KCl, 25 mM KH$_2$PO$_4$, 25 mM glucose, 25 mM EDTA, pH 7.6) respectively, and incubated at 37 °C for 10 min. Subsequently, 1 µL of Benzonase endonuclease (Millipore) was added and the sample gently disaggregated by pipetting and incubated for 5 min at 37 °C. The trypsin was inactivated by adding 1 mL of PBS/5% v/v fetal bovine serum and centrifuged at 1000 x g for 10 min. The sample was resuspended in 100 µL of anti-SSEA1 conjugated to Alexa Fluor 647 (catalog no. MC-480; Biolegend) diluted 1:100 in PBS/2.5% v/v fetal bovine serum and incubated for 10 min at room temperature. Samples were diluted by adding 300 µL of PBS/2.5% v/v fetal bovine serum and passed through a 70 µm filter. For quantification, 300 µL of the samples were immediately run on an ECLIPSE analyzer (Sony Biotechnology) and the data analyzed using FlowJo v10. For sorting of cells, samples were immediately run on a Synergy cell sorter (Sony Biotechnology), the cells sorted into 10 µL of PBS, centrifuged at 3500 x g for 5 min and stored at −80 °C until further analysis.

### Gene expression analysis

The expression of *Rev1, Rev7, Rev3l, Polk* and *Polq* in PGCs was assessed by droplet digital PCR (ddPCR). PGCs (GOF18-GFP$^+$SSEA1$^+$) and somatic cells (GOF18-GFP$^-$SSEA1$^-$) were isolated by FACS as described above. Total RNA was extracted using the PicoPure RNA Isolation Kit (Thermo Fisher Scientific) and first-strand complementary DNA was synthesized using the SuperScript IV Reverse Transcriptase (Thermo Fisher Scientific) according to the manufacturer's instructions. ddPCR was performed using ddPCR Supermix for Probes (BioRad) on a QX ONE Droplet Digital PCR (BioRad) following the manufacturer's instructions. Taqman probes were purchased from ThermoFisher; Rev7, Rev3l, Rev1, Polk, Polq labeled with FAM and Gapdh with VIC. The expression was normalized to *Gapdh* and made relative to the somatic cells. The expression of *Rev1* in MEFs was assessed by purifying RNA with the RNeasy kit (Qiagen) according to the manufacturer's instructions and first-strand complementary DNA was synthesized using the SuperScript IV Reverse Transcriptase (Thermo Fisher Scientific) according to the manufacturer's instructions. PCR amplification was performed using the TaqMan Fast Advanced Master Mix (Thermo Fisher Scientific). PCR amplification was performed on a ViiA 7 cycler for 95 °C for 15 s and 60 °C for 1 min. Mean threshold cycles were determined from three technical repeats using the comparative CT methodology. All expression levels were normalized to *Gapdh* (mm99999915_g1). To perform gene expression analysis on *Rev1$^{-/-}$* and *Pcna$^{R/R}$* PGCs, 50 cells were sorted into 10 µL of Single cell lysis buffer supplemented with DNAseI (Single Cell-to-CT Kit ThermoFisher, 4458237) and stored at −80 °C. The lysis and reverse transcription was performed using the single cell-to-CT kit (ThermoFisher, 4458237) according to the manufacturer's description. Pre-amplification was performed with Taqman Probes (Thermofisher) targeting *Ddx4, Nanos3, Prdm1* and *Gapdh*. PCR amplification was performed using the TaqMan Fast Advanced Master Mix (Thermo Fisher Scientific). PCR amplification was performed on a ViiA 7 cycler for 95 °C for 15 s and 60 °C for 1 min. Mean threshold cycles were determined from three technical repeats using the comparative C$_T$

methodology. All expression levels were normalized to *Gapdh*. For expression analysis of *Dnmt1, Dnmt3a, Dnmt3b, Tet1, Tet2, Stella, Fragilis, Dazl, Mael, Mili, Sycp3, Mov10l1, Dazl, Hormad1* and *Brdt* E12.5 wildtype and *Pcna$^{R/R}$* PGCs were FACS purified as described above. Subsequently, RNA was prepared using the NEBNext® Single Cell/Low Input RNA Library Prep Kit for Illumina® kit and NEBNext® Poly(a) mRNA Magnetic Isolation module kit following the manufacturer's instructions. Subsequently, all libraries were ligated to NEBNext® Multiplex Oligos for Illumina® (Dual Index Primer Set I) and quality control was performed using a 2100 Bioanalyser High Sensitivity DNA Kit 5067-4626 (Agilent) and libraries quantified using a Qubit™ Fluorometer following the manufacturer's instructions. Finally, RNA sequencing libraries were pooled to a final concentration of 8.5 nM. RNA expression was performed using Brilliant II SYBR Green QPCR Master Mix (Agilent) using a ViiA 7 Real-Time PCR system (Thermo Fisher Scientific) at 95 °C for 10 min and 40 cycles of 95 °C for 15 s and 60 °C for 1 min. Mean threshold were determined from three technical repeats per sample and oligonucleotide pair using standard comparative C$_T$ methods. All expression levels were normalized to *Gapdh*. For the testis specific genes *Mov10l1* and *Brdt*, gene expression values were only calculated for male samples.

### Bisulfite sequencing

PGCs (GOF18-GFP$^+$SSEA1$^+$) and somatic cells (GOF18-GFP$^-$SSEA1$^-$) were isolated from E12.5 embryos carrying the GOF18-GFP reporter by FACS as described above. Genomic DNA extraction and sodium bisulfite conversion was performed using the EZ DNA Methylation-Direct Kit (Zymo Research Cat. No. D5020) following the manufacturer's instructions. Nested oligonucleotide pairs described previously were used to amplify *Line-1* (6), *Dazl* (68) and *Mili* (68) sequences using the ZymoTaq polymerase (Zymo Research). PCR products were separated out on an agarose gel and gel-purified using the QIAquick Gel Extraction Kit (Qiagen) following the manufacturer's instructions. Gel extracted products were ligated into the pGEM-T Easy Vector System I (Promega). Ligation products were transformed into *E. coli* and single clones picked following blue-white selection and sent for sequencing using M13R primer. Sequencing reads were analyzed using Quantification for Methylation Analysis (QUMA) software with standard quality control settings.

### Whole genome Bisulfite sequencing

PGCs (GOF18-GFP$^+$SSEA1$^+$) and somatic cells (GOF18-GFP$^-$SSEA1$^-$) were isolated from E12.5 embryos carrying the GOF18-GFP reporter by FACS as described above. WGBS-sequencing libraries were prepared according to the bulk version of the published scBS-seq protocol[109]. Briefly, cells were lysed in RLT plus buffer (Qiagen) then bisulfite converted using the Zymo EZ-Direct kit. Converted DNA was prepared for Illumina sequencing via two rounds of random primed synthesis using oligos containing Illumina adapter sequences followed by PCR to amplify and incorporate indexes. Sequencing was performed on a NextSeq 500 instrument using 75 bp paired end reads. Bismark v0.23.1 was used to align DNA reads to the bisulfite converted GRCm38 mouse genome then perform methylation calling[110]. E6.5 Epiblast data was taken from our previous work selecting a random subset of 50 E6.5 Epiblast cells[111]. For analysis of genomic features (e.g., promoters, gene bodies, 5Kbp tiles), CpG methylation rates were computed assuming a binomial model as previously[111]. To generate the Circos plot, DNA-methylation bigwig files were read into python using pyBigWig library and resampled to 5 Mbp bins. The chromosome band data is obtained from the UCSC table "cytoBandIdeo" of the GRCm38/mm10 assembly. The Circos plot was generated using the pycircos library.

### EdU incorporation analysis in vivo

To assess the in vivo incorporation of 5-ethynyl-2′-deoxyuridine (EdU) into the DNA of PGCs (GOF18-GFP$^+$) the Click-iT™ Plus EdU Cell

Proliferation Kit for Imaging, Alexa Fluor™ 594 kit (Catalog number: C10639) was used. Pregnant mice were given a single dose of EdU (50 mg.kg$^{-1}$) by intraperitoneal injection (IP) at 10 ml.kg$^{-1}$. Females were subsequently culled 4 h post-IP at E12.5 and the embryos harvested; the fetal gonads were dissected and placed into ice-cold PBS. Fetal gonads were fixed in PBS, 4% w/v paraformaldehyde for 30 min at 4 °C. Fixed samples were washed once in PBS for 5 min and then three times in PBS, 1% w/v Triton X-100 for 15 min at room temperature. Samples were then pressed onto glass slides and a large hydrophobic ring drawn around the sample before being incubated in blocking buffer (PBS, 1% w/v Triton X-100, 1% w/v BSA) for 1 h at room temperature. Samples were then incubated with anti-GFP (1:500, catalog no. GF090R; Nacalai) diluted in blocking buffer overnight at 4 °C. Subsequently, samples were washed three times in PBS, 1% w/v Triton X-100 for 5 min at room temperature and fixed in PBS, 2% w/v paraformaldehyde for 20 min at room temperature. Slides were washed three times in PBS and incubated in 200 µL of Click-iT® Plus reaction cocktail made per manufacturer's instructions. Samples were washed three times in PBS, 1% w/v Triton X-100 and incubated with goat anti-rat Alexa Fluor 488 (1:1000, catalog no. A11029, Thermo Fisher Scientific) secondary antibody. Slides were washed three times in PBS, 1% w/v Triton X-100 for 5 min and stained with DAPI (0.5 µg.mL$^{-1}$, PBS) before being mounted with ProLong Gold Antifade Mountant (catalog no. P36934; Thermo Fisher Scientific). Coverslips were placed on top of the slides and allowed to cure for 48 h before images were captured with an LSM780 confocal microscope (Zeiss) and the frequency of positive cells scored blindly.

### Blood counts

A 50 µL total blood sample was taken from saphenous veins or via cardiac puncture and transferred into a K3EDTA MiniCollect tubes (Greiner bio-one) and analysed on a VetABC analyzer, using standard settings for mice (Horiba).

### Serum hormone analysis

For the determination of serum luteinizing hormone, follicle stimulating hormone (FSH) and testosterone levels, blood samples were collected as described above and transferred into a Microvette collection tube (SARSTEDT) and centrifuged at 10,000 x g. for 10 min at room temperature. The supernatant (serum) was transferred to a 1.5 mL eppendorf tube and stored at −80 °C until further analysis. For serum LH and FSH concentrations were determined using the Milliplex Map Mouse Pituitary Magnetic Bead Panel (catalog no. MPTMAG-49K). Plates were loaded manually and washed using a Bio-Plex Pro wash station (BioRad) and data analysis performed using a Magpix Multiplexer reader (BioRad). Serum testosterone levels were determined using the Demeditec Diagnostics rat/mouse ELISA kit (catalog no. DEV9911). Samples were loaded manually and washes performed using a WellWash Versa platewasher (Thermo Scientific). Absorbance was measured at 450 nm using Perkin Elmer Multicalc software. Serum levels of urea, creatinine, aspartate aminotransferase and albumin were measured using a Siemens Dimension RxL analyser.

### Haematopoiesis analysis

Flow cytometry was performed on bone marrow cells that were isolated from the femora and tibiae of mutant mice and appropriate controls by flushing cells and passing them through a 70-µm filter. The following antibodies were used to stain for HSCs: FITC-conjugated lineage cocktail with antibodies anti-CD4 (clone H129.19, BD Pharmingen), CD3e (clone 145-2C11, eBioscience), Ly-6G/Gr-1 (clone RB6-8C5, eBioscience), CD11b/Mac-1 (clone M1/70, BD Pharmingen), CD45R/B220 (clone RA3-6B2, BD Pharmingen), Fcε R1α (clone MAR-1, eBioscience), CD8a (clone 53-6.7, BD Pharmingen), CD11c (clone N418, eBioscience) and TER-119 (clone

Ter119, BD Pharmingen), anti-c-Kit (PerCP-Cy5.5, clone 2B8, eBioscience), anti-Sca-1 (PE-Cy7, clone D7, eBioscience). When staining for SLAM markers the same lineage cocktail was used (FITC) with the addition of the following antibodies: anti-CD48 (FITC, clone HM48-1, BioLegend), anti-CD41 (FITC, clone MWReg30, BD Pharmigen), anti-CD150 (APC, clone TC15-12F12.2, BioLegend) and anti-c-Kit and Sca-1 as above. Maturation of B cells was assessed using anti-CD45R/B220 (PE, clone RA3-6B2, BD Pharmingen) and anti-IgM (APC, clone II/41, BD Pharmingen). The maturation of the erythroid lineage was analysed using antibodies anti-TER-119 (APC, clone Ter-119, BD Pharmingen) and anti-CD71 (PE, clone C2, BD Pharmingen). Granulocyte–macrophage maturation was assessed with antibodies anti-CD11b/Mac-1 (APC, clone M1/70, BD Pharmingen) and anti-Ly-6G/Gr-1 (PE, clone RB6-8C5, eBioscience). Thymic T-cell maturation was assessed using CD4 (FITC, clone H129.19, BD Pharmingen) and CD8a (PE, clone 53-6.7, BD Pharmingen) antibodies. The samples were incubated for 15 min at 4 °C in the dark with the exception of samples containing anti-CD34 (RAM34), which were incubated for 90 min. Samples were run on a LSRII flow cytometer (BD Pharmingen) and the data were analysed with FlowJo v10.

### Micronucleus assay

The micronucleus assay[112,113] was performed by bleeding mice; 62 µl blood was mixed with 338 µl PBS supplemented with 1000 U ml$^{-1}$ of heparin (Calbiochem). 360 µl of blood suspension was then added to 3.6 ml of methanol at −80 °C and stored at −80 °C for at least 12 h. 1 ml of fixed blood cells was then washed with 6 ml of bicarbonate buffer (0.9% NaCl, 5.3 mM NaHCO3). The cells were resuspended in 150 µl of bicarbonate buffer and 20 µl of this suspension was used for subsequent staining. 72 µl of bicarbonate buffer, 1 µl of FITC-conjugated CD71 antibody (GenTex, clone R17217.1.4) and 7 µl RNase A (Sigma) were premixed and added to 20 µl of each cell suspension. The cells were stained at 4 °C for 45 min, followed by addition of 1 ml bicarbonate buffer and centrifugation. Finally, cell pellets were resuspended in 500 µl bicarbonate buffer supplemented with 5 µg ml$^{-1}$ propidium iodide (Sigma). The samples were analysed immediately on an LSRII FACS analyser (BD) and the data analysed with FlowJo v10.

### Statistics and reproducibility

The number of independent biological samples and technical repeats ($n$) are indicated in figure legends. Unless otherwise stated, data are shown as mean ± standard deviation (s.d.) and the nonparametric two-tailed Mann–Whitney test was employed to determine statistical significance. Analysis was performed in GraphPad Prism version 8. The images in Figs. 1l and 3i are representative of the following number of observations (Fig. 1l: Wildtype −7, *Rev1$^{-/-}$* - 12; Fig. 3i: Wildtype – 18, *Pcna$^{R/R}$* – 21).

### Reporting summary

Further information on research design is available in the Nature Portfolio Reporting Summary linked to this article.

## Data availability

The BS-Seq data generated in this study has been deposited in the Gene Expression Omnibus database under accession code GSE253991. The BS-Seq data for E6.5 epiblast used in this study has been previously deposited in the Gene Expression Omnibus database under accession code GSE121708. Source data has been provided with this paper as a Source Data File. All materials are available on request from the corresponding authors. Source data are provided with this paper.

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

## Acknowledgements

We thank A. Surani for the gift of Stella-GFP mice. The hiPSC cell line (BTAG 585b1–868) was a kind gift from Prof. Mitinori Saitou (Institute for the Advanced Study of Human Biology, Kyoto, Japan). We thank the Human Research Tissue Bank (National Institute for Health Research Cambridge Biomedical Research Centre) for processing the histology. We would like to thank Keith Burling and Peter Barker of the NIHR Cambridge Biomedical Research Centre Core Biochemical Assay Laboratory for carrying out the biochemical assays for LH, FSH, albumin, AST, creatinine and urea. Funding was provided by the Medical Research Council as part of UK Research and Innovation file reference no. MC_UP_1201/18 (G.P.C.), Wellcome Trust Investigator Award 210754/Z/18/Z (W.R.), MRC core funding MC-A652-5QA10 (H.G.L. and C.D.), and NIHR Imperial BRC (H.G.L.), Hubrecht Institute (J.I.G.), and the Landsteiner Foundation for Blood Transfusion Research (LSBR) project 2107 (J.W.).

## Author contributions

G.P.C. and P.S. conceived the study, designed experiments and wrote the manuscript. P.S. and R.J.H. designed and performed all experiments and wrote the manuscript. C.D. generated mutant hiPSCs and performed hPGCLC assays. S.J.C. performed bioinformatic analysis of WGBS data. A.A. performed library prep for WGBS. M.J.A. analyzed pathological features of histology samples. J.W. assisted in the micronucleus assay. J.I.G. generated mutant mouse alleles, provided biological samples from various TLS-deficient mouse lines, performed and analysed micronucleus assays, helped design experiments and write the manuscript. H.L. and W.R. helped design experiments and write the manuscript.

## Competing interests

S.J.C., A.A. and W.R. are employees of Altos Labs. W.R. is a consultant and shareholder at Biomodal. The remaining authors declare no competing interests.
