## [Peer Review File · Nature Communications]

Primordial germ cell DNA demethylation and development require DNA translesion synthesisREVIEWER COMMENTS

Reviewer #1 (Remarks to the Author):

This paper from Gerry Crossan and colleagues investigates the requirements for translesion DNA synthesis (TLS) in primordial germ cells (PGCs). The authors target multiple aspects of the TLS machinery, including PCNA, REV1 and several TLS Pols, and provide strong evidence in both human and mouse systems that TLS functions are crucial for PGC population expansion and survival. These findings are consistent with prior reports that the Fanconi Anemia pathway and other elements of the DDR are important in PGCs and extend the findings through thorough analyses and identification of new players in the response. The authors also conduct an in-depth assessment of PGC gene expression and DNA methylation and show for the first time that the defective PGCs fail to undergo normal developmental progression and do not acquire gene expression or methylation patterns typical of later stage WT PGCs. Overall the work highlights important germline-specific requirements for TLS to avoid DNA damage accumulation and enable PGC expansion and survival. The data address an area of interest and are of high quality, but several issues should be addressed prior to publication.

Major points

The abstract states 'It remains unclear if the role of DDR is solely in meiosis' but it is already well established that the DDR is important for PGC expansion and homeostasis.

The paper starts with analysis of REV1 KO iPSC and hPGCLCs, and provides convincing evidence of significant defects in hPGCLC induction/maintenance. Since the remainder of the paper argues for largely germline-specific roles for TLS under normal conditions, it would be useful for the authors to describe if the KO iPSC have any apparent phenotypes under standard conditions (proliferation defects? genomic instability?) Supplementary Figure 1c shows images of the cells after MMC but doesn't show untreated cells, which really is necessary to be able to interpret the data and to understand if there are phenotypes under normal conditions without exogenous damage. Also, is differentiation into other lineages impacted by REV1 loss? Do the KO cells act similarly to WT in teratoma formation assays?

The authors make two different Rev1 alleles to distinguish roles for REV1 catalysis and scaffolding. The scaffolding function is disrupted via a large 100 aa C-terminal deletion, and the resulting mutant (CT) behaves like a null. The authors show that the mutant RNA is expressed but don't provide any evidence that stable protein is generated. Since the deletion could impact protein folding and/or stability, it will be important to show that stable mutant protein is expressed at WT levels or that some functionality is retained to support the conclusion that the CT allele encodes a separation of function mutant and is not just a null allele.

The authors analyze Polk, Polq, and Rev7 mutants. The methods section indicates that the Polk and Polq alleles were described previously. The published reports should be cited in the main text and methods, and the results section should clearly state these are previously reported and validated genetic models. This reviewer can't find where the Rev7 mutant is mentioned in the methods or cited.

The authors compare impacts of the TLS defects in germline vs soma throughout the paper. One useful readout for genomic instability in vivo is measurement of micronucleus formation in peripheral red blood cells, which can be quantified via FACS. Use of this assay would help clarify the extent which these pathways suppress spontaneous genomic instability in vivo, outside of the germline.

To evaluate DNA damage levels the authors stain for gamma-H2AX. Although this is a useful marker, the paper would be strengthened by analysis of additional markers (eg. pCHK1 and pCHK2, among many possibilities) that not only could support the conclusions but might shed light on the nature of the damage that is inducing DDR signaling. The other marker used is RPA, but RPA also coats ssDNA during replication and more clarification is needed as to whether the signal is associated with replication or damage accumulation. Finally, the images in Fig 4a and 4b don't seem to reflect the quantitative data particularly well. In a, the WT and Rev1KO look quite

similarly. In b, the wt and mutants all look similar.

For greater transparency and to facilitate rigorous data assessment, the bar graphs throughout should show the individual data points, not just solid bars.

There are aspects of the discussion that are underdeveloped. For instance, cell cycle accumulation by the mutants in G2 and S is interesting and merits more comment. What does it say about when the damage is occurring during the cell cycle or the checkpoints that respond to it? Since PGCs and ESC share some biological features, how does this cell cycle behavior compare to what is known about ESC cell cycle-specific responses to damage and lack of G1 checkpoint function?

With respect to PGC apoptosis and cell cycle perturbations, some prior studies with DDR mutants have suggested PGC loss is mostly linked to cell cycle arrest, with little or no contributions from apoptosis (eg see PMID: 26388201, 25010009, 5969748). The apoptosis data here are clear, but some discussion relative to the prior studies is warranted.

One of the most fascinating questions remaining is what is the source of the spontaneous DNA lesions in the germline that underlies the dependency on TLS. It is surprising that the authors do not cite or discuss recent work on transcription-replication conflicts in PGCs (PMID: 35969748, 37580696). This topic of the nature of the damage necessitating PGC DDR functions merits further consideration by the authors. The utilization of TLS in the germline also is interesting, since the mechanisms often are viewed as error prone but can correct some damage in an accurate manner. It would be interesting to hear what the authors think about how the reliance on this pathway aligns with low germline mutation rates.

Minor points:

Line 385, GCI presumably should read CGI (although neither abbreviation is ever defined). The sentence wording also is confusing – should it be CGI in promoters of genes... ?

The legend for Fig 3g refers to ovaries and testes, but only ovaries are shown.

There are multiple issues with Figure S1e. The description in the legend (left, right descriptors) doesn't match what is shown. The legend also refers to median values and interquartile ranges, but that is not the format of the data in the figure. The bar graphs lack x axis labels. The legend states 'n=150 tubules per genotype, 50 per genotype'. Presumably that should state 50 per mouse (?). However, for the Rev1 KO genotype there are values only for 100 tubules total. These issues should be corrected.

The legend for Fig. S2 incorrectly refers to panel f as panel e.

Fig S4e graph lacks a y axis label.

Fig S4f y axis label is not accurate (should be GFP+, not GOF18-GFP+).

Fig S4h y axis label lacks numerical values. This reviewer suggests to use the same scale for 4g-i, so that the expansion of PGC numbers per embryo in WT from E9.5 to E11.5 is easier to assess.

Fig. S11 does not appear to be referenced in the text.

Reviewer #2 (Remarks to the Author):

Using multiple approaches and mouse lines, Shah et al. provide compelling evidence that trans-lesion synthesis (TLS) plays a unique and potent role in expansion of primordial germ cells (PGCs) in differentiating human cells and mouse embryos. Initial specification of PGCs is intact, but there is a severe and selective failure to replicate and divide in these cells, with instead cell cycle arrest and apoptosis as outcomes. This is associated with accumulation of DNA damage foci in PGC but

not surrounding somatic tissue. The effects are pre-meiotic as they occur prior to meiotic cell divisions and gamete formation. Likely as a consequence but also possibly entangled with this failure to expand, there is a loss of the normal genome-wide demethylation of induction of germ cell-specific gene expression programs. The data in the manuscript are compelling, with a high level of thoroughness. For example, the PCNA experiments indicate that DNA synthesis through replication-blocking lesions is the problem. A number of extensive and important controls are provided, including analyses of surrounding somatic cells, and even rapidly proliferating hematopoietic stem cells. What is missing from the manuscript is any sense of why this selectively happens in PGCs. The additional experiments in Fig. 5 simply show what would logically be expected given that PGCs don't expand, demethylate and induce germ cell gene expression programs. My other comments are all comparatively minor.

Major Comment:

1. The discussion speculates about the nature of the selective effects on PGC expansion. This mechanism is left unanswered. Any additional data the authors can generate pointing to a mechanism would be the single most important thing that would improve the manuscript.

Minor Comments:

1. Line 97 and again lines 112-114, refs 20-22: these studies all appear to focus on other forms of DNA repair. There is an attempt made in multiple places in the manuscript to link these prior findings to the TLS findings presented here, with terminology such as "factors that deal with DNA damage", but the association as currently presented is tenuous. If there is no prior direct connection, this may actually highlight the novelty of the results, but the text should be clear on whether or not there are prior findings in humans or mice associating TLS with PGCs. Line 116 directly states "TLS factors have been implicated as determinants of human fertility", but there is no provided reference.
2. Line 138: Fig.s 3f and 1g are called out in relation to testes mass, but panel g pertains to ovaries, not testes. Conversely, the lack of difference with ovaries in this panel is not mentioned in the text.
3. Line 174: missing references following "...in line with previous reports."
4. Line 244: the callout for Fig. 3l is instead written as 3i.
5. Figure 4: the nature of the cell-cycle arrest (late G2/early M) could be defined more clearly. In line 333, the description of "abnormal cell cycle kinetics" is a little vague. If it is messy, and the cells are not arresting at a specific point but rather in both late G2 and early M, it should be spelled out clearly.
6. Line 333-335: the wording is awkward, essentially indicating that DNA demethylation of promoters is linked to DNA demethylation...
7. Line 385: The term GCI promoters left undefined. Even more minor point slightly further on: germline responsive genes are defined as "GRRs" on line 392, but in line 395 there is reference to "GRR genes", which would be redundant if GRR indeed stands for germline responsive genes.
8. The findings may suggest that global error-prone repair of DNA lesions is the minor function for these enzymes, with the activity described here for primordia germ cells being the major, and thus perhaps more ancestral, function. A question of pure curiosity: do TLS enzymes, or specific components/protein domains such as in PCNA, appear in the evolutionary record when PGCs in developing embryos are first set aside in this manner? My understanding is that teleosts lack this inductive mechanism of PGC specification and use a different mechanism. Perhaps an *in silico* analysis of the zebrafish proteins would reveal something interesting.

Reviewer #3 (Remarks to the Author):

Crossan-NCOMMS-23-27611-T

The paper by Shah et al. described the role of subunits of the translesion synthesis (TLS) DNA polymerase, Pol zeta, as well as the K164 ubiquitylation of PCNA in the maintenance of primordial germ cells (PGCs), which is a kind of stem cells essential for gamete formation in both male and female. Together with *in vitro* differentiation system of human iPS cells into the primordial germ cell-like cells (PGCLCs), the authors analyzed the PGC cells in mouse embryos of various Rev1

mutants (catalytic dead and C-terminal deletion), Rev7 deletion, and PCNA-K164R mutants and nicely showed that these mutants markedly reduce PGCs. The loss of PGCs in the mutant mice would be induced by the apoptosis and/or cell cycle arrest at the G2/M phase. Importantly, the PGCs in the mutants contained elevated levels of CpG methylation at a global level relative to the wild-type PGCs which showed marked demethylation. This paper contains a large amount of the results with various KO mice, working with PGCs, which are tough to work on. The experiments were performed with great care and the quality of the data in the paper is very high and very much convincing. I have a few points, which would be modified prior to the publication.

Major points:

1. Although the authors nicely showed that a DNA event with the TLS polymerase zeta is critical for the maintenance of the PGCs, previous studies including the characterization of the PGCs in the KO mice of the Rev7, a TLS subunit gene (Watanabe et al. JBC, 2013), which was analyzed in this paper, clearly demonstrated a role of the TLS polymerase in the PGC maintenance. This should be clearly mentioned in the Introduction and the Results sections. It seems that the discoveries of the starting point of the research described in this paper.
2. The authors strangely did not mention about the catalytic subunit of the TLS DNA polymerase zeta, Rev3L, which forms a complex with Rev1 and Rev7. Since the Rev3L KO mice are embryonic lethal (Esposito CB, 2000), it is hard to the role of Rev3L in the PGC development. However, the authors should mention or analyze whether the expression of Rev3L is increased in germ cells or not like other genes such as Rev1 and Rev7. And, since it is very natural to think about the role of Pol zeta in PGC development, the authors should discuss the role of this subunit and Pol zeta clearly rather than talking about the redundancy of the TLS polymerases (Lines 446-449). For example, the Pol zeta is required for DNA synthesis in crosslink repair.
3. The authors' group previously showed the role of Fanconi anemia proteins for DNA crosslink repair such as Ercc1 and Fanca are critical for PGC development (Hills & Crossan, Nature Genet. 2019). It is very important to discuss the relationship between the TLS pathway described here and Fanc proteins (maybe MCM8-9). In the current version, the description on it (lines 471-475) is not enough. This does not mean the analysis of the double mutant such as rev1 fanc1, although the analysis is the best. Given that a TLS mutant such as Rev7 is reported to show a Fanconi anemia-like phenotype (Bluteau et al. 2016, PMID: 27500492), it is critical to check whether the Rev1, Rev7, and PcnA-K164R mutant cells lines are sensitive to the DNA crosslinkers.

Minor points:

1. Introduction: Please describe the role of other DNA repair factors involved in the PGC development, which have been identified by the authors' group and others
2. Line 84: DDR and cell cycle checkpoint are conceptionally similar. Either word is enough here. Or just replace "and" with "or".
3. Line 347, previous data: Please add a reference for the previous data.
4. Figure 4b, line 301-305, RPA staining: Please mention which RPA subunit was stained here. The RPA staining described here is not typical for DNA damage-associated staining of RPA, although the authors called this staining as RPA foci (line 304). RPA associated with DNA damage such as DNA double-strand breaks shows punctate staining called foci. This pan-nuclear staining of RPA corresponds with nuclei in the S-phase. Indeed, the same staining is seen in wild-type control. Also the P-values are >0.05, indicating little difference between WT and mutants.
5. Figure 5I, line 414-417: How about de novo DNA methyl-transferase such as Dnmrt3a, b and Dnmrt3L and other Tet family proteins? It would be nice to mention about the expression of these genes/proteins in the mutant used here. The immuno-staining analysis of Tet1 and Dnmrt1 should be shown.
6. All the graphs should show the data point according to Nature's guidelines. Moreover, the dynamite plot without show a full error bar should be avoided.

REVIEWER COMMENTS

Reviewer #1 (Remarks to the Author):

This paper from Gerry Crossan and colleagues investigates the requirements for translesion DNA synthesis (TLS) in primordial germ cells (PGCs). The authors target multiple aspects of the TLS machinery, including PCNA, REV1 and several TLS Pols, and provide strong evidence in both human and mouse systems that TLS functions are crucial for PGC population expansion and survival. These findings are consistent with prior reports that the Fanconi Anemia pathway and other elements of the DDR are important in PGCs and extend the findings through thorough analyses and identification of new players in the response. The authors also conduct an in-depth assessment of PGC gene expression and DNA methylation and show for the first time that the defective PGCs fail to undergo normal developmental progression and do not acquire gene expression or methylation patterns typical of later stage WT PGCs. Overall the work highlights important germline-specific requirements for TLS to avoid DNA damage accumulation and enable PGC expansion and survival. The data address an area of interest and are of high quality, but several issues should be addressed prior to publication.

Major points

The abstract states 'It remains unclear if the role of DDR is solely in meiosis' but it is already well established that the DDR is important for PGC expansion and homeostasis.

Our Response: We thank the reviewer for raising this issue we have now re-written the sentence to read "The DDR is required during germ cell development and meiosis"

The paper starts with analysis of REV1 KO iPSC and hPGCLCs, and provides convincing evidence of significant defects in hPGCLC induction/maintenance. Since the remainder of the paper argues for largely germline-specific roles for TLS under normal conditions, it would be useful for the authors to describe if the KO iPSC have any apparent phenotypes under standard conditions (proliferation defects? genomic instability?) Supplementary Figure 1c shows images of the cells after MMC but doesn't show untreated cells, which really is necessary to be able to interpret the data and to understand if there are phenotypes under normal conditions without exogenous damage.

Our Response: We thank the reviewer for highlighting the potential that loss of REV1 could lead to a phenotype in iPSCs. In order to test this directly we performed proliferation assays on each of the three REV1-deficient iPSC clones. We found that all three showed comparable proliferation to the parental wildtype iPSC line (**Extended Data Fig. 1d**). We have also included pictures of untreated *REV1*^{-/-} and wildtype iPSCs next to those treated with MMC (**Extended Data Fig. 1c**). Consistent with the proliferation assay, we find no discernible difference between the mutant and wildtype cells.

Also, is differentiation into other lineages impacted by REV1 loss? Do the KO cells act similarly to WT in teratoma formation assays?

Our Response: We agree with the reviewer that assessing the ability of the *REV1*^{-/-} iPSCs to differentiate into different lineages is important. The consensus article by the International Stem Cell Initiative states that either embryoid body (EB) formation or teratoma formation can be used to assess differentiation capacity and recommends that EB formation 'can be accepted as evidence of pluripotency for purposes of standard cell line characterization'¹. Teratoma formation does have the additional benefit of allowing assessment of malignant potential of cell lines, however this must be weighed against the significant impact on animal welfare of this assay. The primary biological question here is regarding differentiation capacity and we therefore opted to perform EB differentiation. The data is presented in **Extended Data Fig. 1e** and shows that the REV1-deficient iPSC clones can differentiate into all three germ layers, in a manner similar to wildtype controls. As such, we conclude that, in agreement with *in vivo* data from *Rev1*^{-/-} mice, there is a specific defect in germline differentiation.

The authors make two different Rev1 alleles to distinguish roles for REV1 catalysis and scaffolding. The scaffolding function is disrupted via a large 100 aa C-terminal deletion, and the resulting mutant (CT) behaves like a null. The authors show that the mutant RNA is expressed but don't provide any evidence that stable protein is generated. Since the deletion could impact protein folding and/or stability, it will be important to show that

stable mutant protein is expressed at WT levels or that some functionality is retained to support the conclusion that the CT allele encodes a separation of function mutant and is not just a null allele.

Our Response: We thank the reviewer for raising this important issue. The C-terminal deletion used in this study is comparable to the deletion used previously². Unfortunately, there are no commercially available antibodies that detect endogenous mouse REV1 (mREV1). We therefore attempted to generate anti-sera against recombinant mREV1. This anti-sera was able to detect recombinant mREV1 by ELISA and when overexpressed in *E. coli* but we were unable to detect the endogenous protein by immunoblot (**Rebuttal Figure 1**).

Rebuttal Figure 1: Generation of an anti-mREV1 anti-sera. (a) Epitope tagged fragments of mREV1 for the production of mREV1 anti-sera were expressed in *E. coli*. The fragments were based on previous publications and cloned into pOPTHM and pOPTG for 6xHis-MBP and GST tagging respectively. (b) Coomassie gels showing successful purification of 6xHis-MBP tagged fragments. Purified recombinant Rev1 was used to immunise 2 rabbits. (c) ELISA analysis against HisMBP-Rev1_476 from the sera of immunised rabbits. Top: serum analysis after 28 days. Bottom: serum analysis after 48 days. (d) Coomassie gels showing successful purification of GST tagged REV1 fragment. This protein was used to generate an affinity purification column using the ThermoFisher AminoLink immobilisation kit to purify antibodies from anti-sera. (e) Purified anti-sera was tested against total lysate and the first elution from *E. coli* expressing recombinant GST tagged REV1 fragment. (f) A range of purified anti-sera dilutions and diluted serum was tested against 40 μ g of total lysates from wildtype and *Rev1*^{-/-} mouse embryonic fibroblasts (MEFs). (g) Purified anti-sera was tested with a larger amount of total lysate (100 μ g) from wildtype and *Rev1*^{-/-} MEFs.

In the absence of a method to detect endogenous mREV1, we tagged both full length and the C-terminal truncation mutant of REV1. We used two epitope tags, GFP and FLAG, fused at the N-terminus of REV1 and transiently expressed both in HEK293 cells. We found that both the FLAG tagged full length and C-terminal truncation of REV1 showed comparable levels by immunoblot with no evidence of degradation products (**Extended Data Fig. 2d**). Furthermore, both the GFP fusion of full length and the C-terminal truncation localised to the nucleus (**Extended Data Fig. 2e**). Together, these data suggest that the C-terminal deletion does not affect localisation or protein stability of REV1.

The authors analyze Polk, Polq, and Rev7 mutants. The methods section indicates that the Polk and Polq alleles were described previously. The published reports should be cited in the main text and methods, and the results section should clearly state these are previously reported and validated genetic models. This reviewer can't find where the Rev7 mutant is mentioned in the methods or cited.

Our Response: We thank the reviewer for highlighting these omissions. We have now ensured that the references for previously generated and validated alleles (Rev7, Polk and Polq) are in both the results and in the materials and methods.

The authors compare impacts of the TLS defects in germline vs soma throughout the paper. One useful readout for genomic instability in vivo is measurement of micronucleus formation in peripheral red blood cells, which can be quantified via FACS. Use of this assay would help clarify the extent which these pathways suppress spontaneous genomic instability in vivo, outside of the germline.

Our Response: We thank the reviewer for this suggestion. We have now performed the micronucleus assay on our panel of TLS mutants as described previously³⁻⁵. We found an induction in the frequency of micronucleated normochromic erythrocytes (Mn-NCE) in *Pcna^{R/R}*, *Rev1^{-/-}*, and *Polq^{-/-}* (the induction in *Rev7^{-/-}* was not statistically significant) demonstrating that there is genomic instability in the hematopoietic compartment (**Extended Data Fig. 5c**). Despite this, the maturation and frequency of peripheral blood cells is not affected (**Extended Data Figs. 5d-e and 5g-l**). This is consistent with the observed reduction of HSCs reported previously for *Rev1^{-/-}* mice and reported here for *Pcna^{R/R}* mice which have a small but significant reduction in HSC frequency⁶. These data indicate that these TLS factors do play a role in maintaining genome stability in the hematopoietic compartment. However, it is striking that the reduction in HSC frequency is approximately 2-fold which is substantially less than the >100-fold reduction observed in PGCs. Our study demonstrates that the loss of REV1, REV7 or the PCNA K164R mutation results in increased genome instability in the blood and germ cell compartments but whilst blood production and homeostasis remains intact, there is a complete lack of germ cells into adulthood accompanied by infertility

To evaluate DNA damage levels the authors stain for gamma-H2AX. Although this is a useful marker, the paper would be strengthened by analysis of additional markers (eg. pCHK1 and pCHK2, among many possibilities) that not only could support the conclusions but might shed light on the nature of the damage that is inducing DDR signaling. The other marker used is RPA, but RPA also coats ssDNA during replication and more clarification is needed as to whether the signal is associated with replication or damage accumulation. Finally, the images in Fig 4a and 4b don't seem to reflect the quantitative data particularly well. In a, the WT and Rev1KO look quite similarly. In b, the wt and mutants all look similar.

Our Response: We thank the reviewer for their input. In the previous version the text and figure legend were unclear when describing the RPA data. We stained for the RPA32 subunit and have now stated this clearly. Whilst pan nuclear RPA staining is characteristic of cells in S-phase, foci are associated with DNA damage and in our analysis we quantified the frequency of PGCs containing RPA foci - we have now updated the text, legend and axis to make this clear^{7,8}.

We went on to now assess staining for pCHK2 and the data is presented in **Fig. 4c**. We find that there is a significant increase in the frequency of *Pcna^{R/R}* and *Rev1^{-/-}* PGCs that accumulate pCHK2. We did attempt to stain with pCHK1 antibodies however this was technically unsuccessful, and we were unable to observe a signal even when mice were exposed to an exogenous source of DNA damage and showed a significant induction of γ -H2A.X foci.

Rebuttal Figure 2: Failure to detect pCHK1 by immunofluorescence. E12.5 pregnant females were injected with 0.2 mg.ml^{-1} mitomycin C (MMC) at 20 ml.kg^{-1} to induce DNA damage. 4 hours after MMC exposure, embryos were harvested, and the embryonic gonads dissected. These were stained for pCHK1 (Cell Signalling Technology, 2348), γ -H2A.X as a positive control and DAPI.

We have now included the following sentence in the discussion: “We observe an increase in the frequency of TLS deficient PGCs with γ -H2A.X foci, RPA32 foci and pCHK2. Whilst none of these markers are entirely specific to the accumulation of DNA DSBs the combination of all three strongly suggests that in the absence of TLS PGCs accumulate DNA DSBs⁹.”

For greater transparency and to facilitate rigorous data assessment, the bar graphs throughout should show the individual data points, not just solid bars.

Our Response: We have now updated the figures to match the requirements by the journal and included individual data points throughout the manuscript.

There are aspects of the discussion that are underdeveloped. For instance, cell cycle accumulation by the mutants in G2 and S is interesting and merits more comment. What does it say about when the damage is occurring during the cell cycle or the checkpoints that respond to it? Since PGCs and ESC share some biological features, how does this cell cycle behaviour compare to what is known about ESC cell cycle-specific responses to damage and lack of G1 checkpoint function? With respect to PGC apoptosis and cell cycle perturbations, some prior studies with DDR mutants have suggested PGC loss is mostly linked to cell cycle arrest, with little or no contributions from apoptosis (eg see PMID: 26388201, 25010009, 5969748). The apoptosis data here are clear, but some discussion relative to the prior studies is warranted.

Our Response: We thank the reviewer for this comment. We agree that this is a particularly interesting aspect of our study. We have now restructured a paragraph within the discussion to enable us to discuss the nature of DNA damage in more detail. This is followed by a more thorough discussion of the cell cycle defects and finally we discuss our apoptosis observation in the context of previously published data.

One of the most fascinating questions remaining is what is the source of the spontaneous DNA lesions in the germline that underlies the dependency on TLS. It is surprising that the authors do not cite or discuss recent work on transcription-replication conflicts in PGCs (PMID: 35969748, 37580696). This topic of the nature of the damage necessitating PGC DDR functions merits further consideration by the authors. The utilization of TLS in the germline also is interesting, since the mechanisms often are viewed as error prone but can correct some damage in an accurate manner. It would be interesting to hear what the authors think about how the reliance on this pathway aligns with low germline mutation rates.

Our Response: We thank the reviewer for their comment. We agree that one of the interesting questions is why there is selectivity in the germline and what the nature of the lesion is. This is a difficult problem throughout the field of DNA repair and whilst the nature of insult that many pathways can deal with is understood (e.g. when cells are exposed to genotoxins in tissue culture cells) we have much less confidence/evidence for the source(s) of DNA

damage that these pathways repair *in vivo* under physiological conditions. We have now expanded the discussion and speculate about potential sources of damage that may occur in PGCs (lines 567-575).

We further agree with the reviewer that, on the surface, it appears contradictory that germ cells have lower rates of mutagenesis when compared to somatic cells but have an essential requirement for TLS, which employs mutagenic polymerases¹⁰. Firstly, it is not known what proportion of germline mutations arise from TLS when compared to other mutagenic processes (e.g. the misincorporation of nucleotides during DNA replication). It is plausible that TLS is essential to repair a very infrequent lesion – failure to repair this lesion results in DNA double strand breaks (with cell cycle arrest or apoptosis) and PGC loss. In contrast, when TLS is active it repairs these lesions with a potential mutagenic penalty. However, due to their scarcity these lesions (and their mutagenic repair) may contribute only a very small proportion of *de novo* germline mutations. It is worth noting that the majority of germline mutations can be explained by single base substitution signatures (SBS) 1 and 5¹¹⁻¹⁶. SBS1 is likely caused by the deamination of methylcytosine and therefore unlikely to be driven by TLS¹⁷. The aetiology of SBS5 is not understood but this process is known to also occur in somatic tissues so is not a germ line specific process¹⁸. Therefore, the majority of germ line mutations occur without the action of the TLS machinery.

Minor points:

Line 385, GCI presumably should read CGI (although neither abbreviation is ever defined). The sentence wording also is confusing – should it be CGI in promoters of genes... ?

Our Response: We thank the reviewer for highlighting this error. We have now changed this to read “CGIs (CpG islands) in the promoters of genes associated with germ cell specific processes”.

The legend for Fig 3g refers to ovaries and testes, but only ovaries are shown.

Our Response: We thank the reviewer for highlighting this error and we have now ensured that it refers only to ovaries.

There are multiple issues with Figure S1e. The description in the legend (left, right descriptors) doesn't match what is shown. The legend also refers to median values and interquartile ranges, but that is not the format of the data in the figure. The bar graphs lack x axis labels. The legend states 'n=150 tubules per genotype, 50 per genotype'. Presumably that should state 50 per mouse (?). However, for the Rev1 KO genotype there are values only for 100 tubules total. These issues should be corrected.

Our Response: We thank the reviewer for highlighting this error and this has now been changed so the figure legend accurately describes the figure panel.

The legend for Fig. S2 incorrectly refers to panel f as panel e.

Our Response: We thank the reviewer for highlighting this error and we have now changed this figure legend to call the panel correctly.

Fig S4e graph lacks a y axis label.

Our Response: We thank the reviewer for highlighting this error and we have now added a y-axis label to this graph

Fig S4f y axis label is not accurate (should be GFP+, not GOF18-GFP+).

Our Response: We thank the reviewer for highlighting this error and we have now changed this axis label to say GFP+.

Fig S4h y axis label lacks numerical values. This reviewer suggests to use the same scale for 4g-i, so that the expansion of PGC numbers per embryo in WT from E9.5 to E11.5 is easier to assess.

Our Response: We thank the reviewer for highlighting this error and we have changed all 3 panels of Extended Data Fig. 4g-i to have the same y-axes.

Fig. S11 does not appear to be referenced in the text.

Our Response: We have now removed this model as it added little to the narrative.

Reviewer #2 (Remarks to the Author):

Using multiple approaches and mouse lines, Shah et al. provide compelling evidence that trans-lesion synthesis (TLS) plays a unique and potent role in expansion of primordial germ cells (PGCs) in differentiating human cells and mouse embryos. Initial specification of PGCs is intact, but there is a severe and selective failure to replicate and divide in these cells, with instead cell cycle arrest and apoptosis as outcomes. This is associated with accumulation of DNA damage foci in PGC but not surrounding somatic tissue. The effects are pre-meiotic as they occur prior to meiotic cell divisions and gamete formation. Likely as a consequence but also possibly entangled with this failure to expand, there is a loss of the normal genome-wide demethylation of induction of germ cell-specific gene expression programs. The data in the manuscript are compelling, with a high level of thoroughness. For example, the PCNA experiments indicate that DNA synthesis through replication-blocking lesions is the problem. A number of extensive and important controls are provided, including analyses of surrounding somatic cells, and even rapidly proliferating hematopoietic stem cells. What is missing from the manuscript is any sense of why this selectively happens in PGCs. The additional experiments in Fig. 5 simply show what would logically be expected given that PGCs don't expand, demethylate and induce germ cell gene expression programs. My other comments are all comparatively minor.

Major Comment:

1. The discussion speculates about the nature of the selective effects on PGC expansion. This mechanism is left unanswered. Any additional data the authors can generate pointing to a mechanism would be the single most important thing that would improve the manuscript.

Our Response: We thank the reviewer and are pleased that they find the evidence provided compelling and thorough. We agree that a key question going forward will be to understand the mechanism that leads to the selective effect of TLS loss on PGC development. This manuscript includes a large amount of data generated in cellular models, in vivo, and by in depth phenotyping. We believe that the mechanistic studies will form the basis of future work but we have expanded the discussion to include potential reasons to explain the selectivity of the observed phenotypes.

Minor Comments:

1. Line 97 and again lines 112-114, refs 20-22: these studies all appear to focus on other forms of DNA repair. There is an attempt made in multiple places in the manuscript to link these prior findings to the TLS findings presented here, with terminology such as "factors that deal with DNA damage", but the association as currently presented is tenuous. If there is no prior direct connection, this may actually highlight the novelty of the results, but the text should be clear on whether or not there are prior findings in humans or mice associating TLS with PGCs. Line 116 directly states "TLS factors have been implicated as determinants of human fertility", but there is no provided reference.

Our Response: We thank the reviewer for this response and we agree that the previous version used unclear wording. We have now re-written this paragraph to explicitly draw a distinction between GWAS studies, previous studies in mouse models (Reviewer 3) and the current study.

2. Line 138: Fig.s 3f and 1g are called out in relation to testes mass, but panel g pertains to ovaries, not testes. Conversely, the lack of difference with ovaries in this panel is not mentioned in the text.

Our Response: We thank the reviewer for highlighting this error and we have now corrected the figure call out and explicitly stated that the ovary mass is unaffected.

3. Line 174: missing references following "...in line with previous reports."

Our Response: We thank the reviewer for this comment, and we have now included references to support this statement.

4. Line 244: the callout for Fig. 3I is instead written as 3i.

Our Response: We thank the reviewer for highlighting this error and we have now corrected the callout.

5. Figure 4: the nature of the cell-cycle arrest (late G2/early M) could be defined more clearly. In line 333, the description of "abnormal cell cycle kinetics" is a little vague. If it is messy, and the cells are not arresting at a specific point but rather in both late G2 and early M, it should be spelled out clearly.

Our Response: We thank the reviewer for this comment, and we agree that the previous phrasing was vague. We observed an increase in the frequency of mutant PGCs with nuclear cyclin B1 which is characteristic of cells at the end of G2/early-M phase. In addition, we see an increase in the proportion of TLS-deficient PGCs that stain positive for Ser10 phosphorylation of histone H3 which is characteristic of entry into M-phase. Finally, we see a reduction in the frequency of mutant PGCs that incorporate EdU during a 4-hour pulse, indicating that a reduced frequency of PGCs undergo DNA replication, and hence S-phase, during this window compared to wildtype. Taken together these data are consistent with the accumulation of PGCs at the G2/M phase of the cell cycle in the absence of TLS. We have now removed the phrase "abnormal cell cycle kinetics" and replaced it with "accumulate at the G2/M phase of the cell cycle".

6. Line 333-335: the wording is awkward, essentially indicating that DNA demethylation of promoters is linked to DNA demethylation...

Our Response: We thank the reviewer for this comment, we have now changed the text to make it read clearer.

7. Line 385: The term GCI promoters left undefined. Even more minor point slightly further on: germline responsive genes are defined as "GRRs" on line 392, but in line 395 there is reference to "GRR genes", which would be redundant if GRR indeed stands for germline responsive genes.

Our Response: We thank the reviewer for this comment, and we have defined CGI promoters in the text (GCI was unfortunately a typographical error). We have also corrected the call out of GRRs.

8. The findings may suggest that global error-prone repair of DNA lesions is the minor function for these enzymes, with the activity described here for primordial germ cells being the major, and thus perhaps more ancestral, function. A question of pure curiosity: do TLS enzymes, or specific components/protein domains such as in PCNA, appear in the evolutionary record when PGCs in developing embryos are first set aside in this manner? My understanding is that teleosts lack this inductive mechanism of PGC specification and use a different mechanism. Perhaps an in silico analysis of the zebrafish proteins would reveal something interesting.

Our Response: The process of translesion synthesis and the factors involved are widely distributed among species^{10,19}. REV1 and REV7, the TLS factors that we investigate here, are conserved in budding yeast along with PCNA and its post-translation modification being conserved among eukaryotes²⁰⁻²⁵. There do appear to be differences in the importance of these factors in yeast when compared to vertebrates. In yeast, REV1 is responsible for the majority of spontaneous and induced base substitutions²⁶. Recently, it has shown that REV1 is also required for a substantial proportion of base substitutions in vertebrates, but this is somewhat less striking than in yeast²⁷. Together, these data argue that the central function of TLS in controlling mutagenesis is conserved.

It seems unlikely that the ancestral function of these factors was in PGC specification given that these processes are present in yeast, having arisen prior to organisms employing discontinuous germline propagation.

Reviewer #3 (Remarks to the Author):

Crossan-NCOMMS-23-27611-T

The paper by Shah et al. described the role of subunits of the translesion synthesis (TLS) DNA polymerase, Pol zeta, as well as the K164 ubiquitylation of PCNA in the maintenance of primordial germ cells (PGCs), which is a kind of stem cells essential for gamete formation in both male and female. Together with in vitro differentiation system of human iPS cells into the primordial germ cell-like cells (PGCLCs), the authors analyzed the PGC cells in mouse embryos of various Rev1 mutants (catalytic dead and C-terminal deletion), Rev7 deletion, and PCNA-K164R mutants and nicely showed that these mutants markedly reduce PGCs. The loss of PGCs in the mutant mice would be induced by the apoptosis and/or cell cycle arrest at the G2/M phase. Importantly, the PGCs in the mutants contained elevated levels of CpG methylation at a global level relative to the wild-type PGCs which showed marked demethylation. This paper contains a large amount of the results with various KO mice, working with PGCs, which are tough to work on. The experiments were performed with great care and the quality of the data in the paper is very high and very much convincing. I have a few points, which would be modified prior to the publication.

Major points:

1. Although the authors nicely showed that a DNA event with the TLS polymerase zeta is critical for the maintenance of the PGCs, previous studies including the characterization of the PGCs in the KO mice of the Rev7, a TLS subunit gene (Watanabe et al. JBC, 2013), which was analyzed in this paper, clearly demonstrated a role of the TLS polymerase in the PGC maintenance. This should be clearly mentioned in the Introduction and the Results sections. It seems that the discoveries of the starting point of the research described in this paper.

Our Response: We thank the reviewer for highlighting this and we apologise for this omission. We have re-written the beginning of the results section to make distinctions between GWAS studies and previous animal studies on DNA repair factors and specifically address the previous work surrounding REV7. We highlight the multiple functions of REV7 in DNA repair/mitosis and then go on to state that we will specifically test if there is a role for TLS during PGC development.

2. The authors strangely did not mention about the catalytic subunit of the TLS DNA polymerase zeta, Rev3L, which forms a complex with Rev1 and Rev7. Since the Rev3L KO mice are embryonic lethal (Esposito CB, 2000), it is hard to the role of Rev3L in the PGC development. However, the authors should mention or analyze whether the expression of Rev3L is increased in germ cells or not like other genes such as Rev1 and Rev7.

Our Response: We thank the reviewer for their input on this point. We agree with the assessment that REV7 is likely to act with REV3L in PGC development. As noted by the reviewer this is difficult to test *in vivo* as REV3L is essential for embryonic development before E10.5 and possibly as early as in the blastocyst²⁸⁻³⁰. As suggested by the reviewer we have now assessed the expression of REV3L in PGCs at E10.5 and see a similar pattern of expression as was observed for *Rev1* and *Rev7* (**Extended Data Fig. 3b**). This supports our other findings which suggest that REV7 is acting as part of the Polζ polymerase together with with REV3L during PGC development.

And, since it is very natural to think about the role of Pol zeta in PGC development, the authors should discuss the role of this subunit and Pol zeta clearly rather than talking about the redundancy of the TLS polymerases (Lines 446-449). For example, the Pol zeta is required for DNA synthesis in crosslink repair.

Our Response: We thank the reviewer for highlighting this omission. We have now explicitly discussed the role of REV7 in Polζ (and other repair transactions). We agree with the reviewer that it is very natural to think of the role of REV7 in Polζ in the context of this work. Indeed, we think the requirement for REV7 is due to its role in Polζ. This also fits with the requirement for the C-terminal region of REV1 which interacts and recruits Polζ.

3. The authors' group previously showed the role of Fanconi anemia proteins for DNA crosslink repair such as *Erc1* and *Fanca* are critical for PGC development (Hills & Crossan, Nature Genet. 2019). It is very important to discuss the relationship between the TLS pathway described here and Fanc proteins (maybe MCM8-9). In the current version, the description on it (lines 471-475) is not enough. This does not mean the analysis of the double mutant such as *rev1 fanc1*, although the analysis is the best. Given that a TLS mutant such as *Rev7* is reported to show a Fanconi anemia-like phenotype (Bluteau et al. 2016, PMID: 27500492), it is critical to check whether the *Rev1*, *Rev7*, and *Pcna-K164R* mutant cells lines are sensitive to the DNA crosslinkers.

Our Response: The reviewer highlights an interesting and important issue. Firstly, we find that *Rev1*^{-/-}, *Rev7*^{-/-} and *Pcna*^{R/R} mouse embryonic fibroblasts (MEFs) are mildly hypersensitive to the DNA interstrand crosslinking agent MMC (**Supplementary Fig. 1a**). This is not surprising given previous genetic and biochemical analysis which places these TLS factors downstream of the Fanconi Anaemia DNA interstrand crosslink repair pathway^{31,32}. However, this sensitivity is substantially less than what is observed in *Fanca*^{-/-} MEFs. Furthermore, cells deficient in these TLS factors are hypersensitive to a wide range of different DNA damaging agents that do not necessitate FA interstrand crosslink repair pathway to maintain cellular resistance (e.g. MMS (alkylating agent), UV irradiation (intrastrand crosslink and bulky lesions, 4NQO (purine adducts)^{2,33,34}).

It is worth considering if the PGC defect observed in TLS mutant mice is due to a defect in ICL repair. If this were correct it would be expected that there would be phenotypic overlap between mice deficient in FA interstrand crosslink repair and those deficient in TLS, especially given that TLS acts at the terminal phase of ICL repair. However, whilst both classes of repair mutants have reduced numbers of PGCs by E12.5 there are substantial phenotypic differences showing that it is not the role of TLS factors in ICL repair that explains the phenotype we have uncovered (**Rebuttal Fig. 3**). Firstly, loss of TLS results in a complete sterility whilst loss of FA greatly attenuates fertility (**Rebuttal Fig. 3a**). Therefore whilst we have never observed a TLS deficient mouse give rise to progeny this occurs frequently in FA-deficient mice³⁵. Secondly, the temporality of PGC loss is different between TLS mutants and FA mutants: the loss of PGCs is evident in TLS mutants 48h before any loss is observed in FA deficient embryos (**Rebuttal Fig. 3b**). Thirdly, the magnitude of PGC loss at E12.5 is comparable among all TLS mutants but this is 2 orders of magnitude greater than the loss observed in FA-deficient embryos (**Rebuttal Fig. 3c**). Finally, there are phenotypic differences between TLS- and FA-deficient PGCs. TLS-deficient PGCs have both epigenetic and germ cell factor transcriptional changes (**Rebuttal Fig. 3d**). This is not the case in FA-deficient PGCs in which DNA demethylation is intact as is the activation of the germ cell transcriptional programme. Taken together these data show that whilst both TLS- and FA-deficient mice have PGC defects, the germ cell phenotype observed in embryos lacking TLS factors is distinct from that observed in FA deficiency.

We have now expanded the discussion to explicitly outline the molecular role of TLS factors in FA-crosslink repair and provide a detailed description of the phenotypic differences between the two classes of mutants.

Rebuttal Figure 3: Comparison of germ cell phenotypes in TLS- and FA-deficient organisms. (a) Left: Cumulative number of offspring when wildtype or mutant mice were mated with wildtype mates of the opposite sex. **Right:** Number of offspring born per litter when wildtype or mutant mice were mated with wildtype mates of the opposite sex. **(b) Left:** Frequency of PGCs relative to wildtype during embryonic development in wildtype and mutant embryos. **Right:** Quantification of number of PGCs in E12.5 embryos. **(c) Left:** Representative genomic bisulfite sequencing reads of the CpG-rich region of the *Line-1* element from FACS-purified wildtype somatic cells and from E12.5 wildtype and mutant PGCs (filled = methylated CpG, open = unmethylated CpG). **Right:** Quantification of methylated CpG dinucleotides in the *Line-1* element in wildtype somatic cells and in E12.5 wildtype and mutant PGCs.

Minor points:

1. Introduction: Please describe the role of other DNA repair factors involved in the PGC development, which have been identified by the authors' group and others

Our Response: We have now re-written the text to highlight and include references to previous studies showing a role for DNA repair/mitotic factors during PGC development (lines 113-115).

2. Line 84: DDR and cell cycle checkpoint are conceptually similar. Either word is enough here. Or just replace "and" with "or".

Our Response: We have now changed the text to read "DNA damage response (DDR) including cell cycle checkpoints".

3. Line 347, previous data: Please add a reference for the previous data.

Our Response: We apologise for the confusion caused here. The "previous data" referred to the gene expression data presented in this manuscript for *Nanos3*, *Prdm1* and *Mvh*. For greater clarity, the wording has now been changed to "Consistent with the gene expression data presented above" (line 364-365).

4. Figure 4b, line 301-305, RPA staining: Please mention which RPA subunit was stained here. The RPA staining described here is not typical for DNA damage-associated staining of RPA, although the authors called this staining as RPA foci (line 304). RPA associated with DNA damage such as DNA double-strand breaks shows punctate staining called foci. This pan-nuclear staining of RPA corresponds with nuclei in the S-phase. Indeed, the same staining is seen in wild-type control. Also the P-values are >0.05, indicating little difference between WT and mutants.

Our Response: We thank the reviewer for highlighting these issues. Firstly, we would like to apologise about the confusion that has been caused by the original text and figure legend - both were poorly written and the images chosen did not highlight what we were quantifying. The staining was the RPA32 subunit of RPA and we quantified cells with foci. We have now updated the text and the figure legend to accurately reflect what we actually measured. Secondly, we performed additional biological replicates, so we now have $n > 3$ for all genotypes and when we perform statistical analysis, we find a significant increase in the mutants (*Pcna^{R/R}*; $P = 0.0159$ and *Rev1^{-/-}*; $P = 0.0357$). We have updated the text to reflect this. In addition, we have performed staining for pCHK2 and found a significant increase in the frequency of *Pcna^{R/R}* and *Rev1^{-/-}* PGCs that stain positive for pCHK2 (**Fig. 4c**). This provides a converging line of evidence that TLS-deficient PGCs accumulate DNA damage and activate a DDR.

5. Figure 5l, line 414-417: How about de novo DNA methyl-transferase such as Dnmt3a, b and Dnmt3L and other Tet family proteins? It would be nice to mention about the expression of these genes/proteins in the mutant used here. The immuno-staining analysis of Tet1 and Dnmt1 should be shown.

Our Response: We agree with the reviewer that it is worthwhile investigating the expression of other methyltransferases and TET family members as increased methyltransferase or decreased TET activities may explain the retention of DNA methylation in TLS-deficient PGCs. Previous studies have shown *Dnmt1* to be the only methyltransferase expressed in E12.5 PGCs with a lack of *Dnmt3a* or *Dnmt3b* expression³⁶⁻³⁸. Consistent with this, we found *Dnmt1* to be expressed in both wildtype and *Pcna^{R/R}* PGCs at similar levels but a lack of expression of the *de novo* *Dnmt3a* and *Dnmt3b* methyltransferases in either (**Fig. 5j** and **Extended Data Fig. 9e**). We further went on to assess *Dnmt3l* but did not detect its expression consistent with reports that *Dnmt3l* expression initiates between E13.5-E15.5 in male PGCs^{37,39}. We attempted to stain for DNMT1 protein in PGCs however, we did not observe a clear nuclear signal in wildtype PGCs. However, our mRNA expression analysis is clear and in agreement with published datasets³⁶⁻³⁸. There are three TET family members present in mammals with TET3 being an isoform specific to the zygote/oocyte⁴⁰⁻⁴². Previous work has revealed that of the two other members, TET1 is the only active member in PGCs^{37,40,43,44}. Consistent with this, we did not detect *Tet2* expression in wildtype or *Pcna^{R/R}* PGCs (**Extended Data Fig. 9f**). Furthermore, we did not find a difference in the expression of *Tet1* in *Pcna^{R/R}* PGCs compared to wildtype and immunostaining of E12.5 wildtype and *Pcna^{R/R}* embryonic gonads revealed similar levels of protein (**Extended Data Fig. 9g**). Hence, we do not find evidence to suggest that changes in the expression of either methyltransferases or TET family members could explain the retention in DNA methylation in TLS-deficient PGCs.

6. All the graphs should show the data point according to Nature's guidelines. Moreover, the dynamite plot without show a full error bar should be avoided.

Our Response: We have now updated the figures to match the requirements by the journal and included individual data points throughout the manuscript.

References

1. International Stem Cell, I. Assessment of established techniques to determine developmental and malignant potential of human pluripotent stem cells. *Nat Commun* **9**, 1925 (2018).
2. Ross, A.L., Simpson, L.J. & Sale, J.E. Vertebrate DNA damage tolerance requires the C-terminus but not BRCT or transferase domains of REV1. *Nucleic Acids Res* **33**, 1280-9 (2005).
3. Garaycochea, J.I. *et al.* Alcohol and endogenous aldehydes damage chromosomes and mutate stem cells. *Nature* **553**, 171-177 (2018).
4. Adams, D.J. *et al.* BRCTx is a novel, highly conserved RAD18-interacting protein. *Mol Cell Biol* **25**, 779-88 (2005).
5. Reinholdt, L., Ashley, T., Schimenti, J. & Shima, N. Forward genetic screens for meiotic and mitotic recombination-defective mutants in mice. *Methods Mol Biol* **262**, 87-107 (2004).
6. Pilzecker, B. *et al.* DNA damage tolerance in hematopoietic stem and progenitor cells in mice. *Proc Natl Acad Sci U S A* **114**, E6875-E6883 (2017).
7. Vassin, V.M., Wold, M.S. & Borowiec, J.A. Replication protein A (RPA) phosphorylation prevents RPA association with replication centers. *Mol Cell Biol* **24**, 1930-43 (2004).
8. Golub, E.I., Gupta, R.C., Haaf, T., Wold, M.S. & Radding, C.M. Interaction of human rad51 recombination protein with single-stranded DNA binding protein, RPA. *Nucleic Acids Res* **26**, 5388-93 (1998).
9. Zannini, L., Delia, D. & Buscemi, G. CHK2 kinase in the DNA damage response and beyond. *J Mol Cell Biol* **6**, 442-57 (2014).
10. Sale, J.E., Lehmann, A.R. & Woodgate, R. Y-family DNA polymerases and their role in tolerance of cellular DNA damage. *Nat Rev Mol Cell Biol* **13**, 141-52 (2012).
11. Beichman, A.C. *et al.* Evolution of the Mutation Spectrum Across a Mammalian Phylogeny. *Mol Biol Evol* **40**(2023).
12. Moore, L. *et al.* The mutational landscape of human somatic and germline cells. *Nature* **597**, 381-386 (2021).
13. Lindsay, S.J., Rahbari, R., Kaplanis, J., Keane, T. & Hurles, M.E. Similarities and differences in patterns of germline mutation between mice and humans. *Nat Commun* **10**, 4053 (2019).
14. Rahbari, R. *et al.* Timing, rates and spectra of human germline mutation. *Nat Genet* **48**, 126-133 (2016).
15. Acuna-Hidalgo, R., Veltman, J.A. & Hoischen, A. New insights into the generation and role of de novo mutations in health and disease. *Genome Biol* **17**, 241 (2016).
16. Spisak, N., de Manuel, M., Milligan, W., Sella, G. & Przeworski, M. Disentangling sources of clock-like mutations in germline and soma. *bioRxiv* (2023).
17. Nik-Zainal, S. *et al.* Mutational processes molding the genomes of 21 breast cancers. *Cell* **149**, 979-93 (2012).
18. Alexandrov, L.B. *et al.* The repertoire of mutational signatures in human cancer. *Nature* **578**, 94-101 (2020).
19. Sale, J.E. Translesion DNA synthesis and mutagenesis in eukaryotes. *Cold Spring Harb Perspect Biol* **5**, a012708 (2013).
20. Acharya, N. *et al.* Complex formation of yeast Rev1 and Rev7 proteins: a novel role for the polymerase-associated domain. *Mol Cell Biol* **25**, 9734-40 (2005).
21. Lawrence, C.W., Das, G. & Christensen, R.B. REV7, a new gene concerned with UV mutagenesis in yeast. *Mol Gen Genet* **200**, 80-5 (1985).
22. Hoegge, C., Pfander, B., Moldovan, G.L., Pyrowolakis, G. & Jentsch, S. RAD6-dependent DNA repair is linked to modification of PCNA by ubiquitin and SUMO. *Nature* **419**, 135-41 (2002).

23. Stelter, P. & Ulrich, H.D. Control of spontaneous and damage-induced mutagenesis by SUMO and ubiquitin conjugation. *Nature* **425**, 188-91 (2003).
24. Moldovan, G.L., Pfander, B. & Jentsch, S. PCNA, the maestro of the replication fork. *Cell* **129**, 665-79 (2007).
25. Nelson, J.R., Lawrence, C.W. & Hinkle, D.C. Deoxycytidyl transferase activity of yeast REV1 protein. *Nature* **382**, 729-31 (1996).
26. Lawrence, C.W. Cellular roles of DNA polymerase zeta and Rev1 protein. *DNA Repair (Amst)* **1**, 425-35 (2002).
27. Gyure, Z. *et al.* Spontaneous mutagenesis in human cells is controlled by REV1-Polymerase zeta and PRIMPOL. *Cell Rep* **42**, 112887 (2023).
28. Esposito, G. *et al.* Disruption of the Rev3l-encoded catalytic subunit of polymerase zeta in mice results in early embryonic lethality. *Curr Biol* **10**, 1221-4 (2000).
29. Bemark, M., Khamlichi, A.A., Davies, S.L. & Neuberger, M.S. Disruption of mouse polymerase zeta (Rev3) leads to embryonic lethality and impairs blastocyst development in vitro. *Curr Biol* **10**, 1213-6 (2000).
30. Wittschieben, J. *et al.* Disruption of the developmentally regulated Rev3l gene causes embryonic lethality. *Curr Biol* **10**, 1217-20 (2000).
31. Budzowska, M., Graham, T.G., Sobock, A., Waga, S. & Walter, J.C. Regulation of the Rev1-pol zeta complex during bypass of a DNA interstrand cross-link. *EMBO J* **34**, 1971-85 (2015).
32. Niedzwiedz, W. *et al.* The Fanconi anaemia gene FANCC promotes homologous recombination and error-prone DNA repair. *Mol Cell* **15**, 607-20 (2004).
33. Simpson, L.J. & Sale, J.E. Rev1 is essential for DNA damage tolerance and non-templated immunoglobulin gene mutation in a vertebrate cell line. *EMBO J* **22**, 1654-64 (2003).
34. Guo, C. *et al.* REV1 protein interacts with PCNA: significance of the REV1 BRCT domain in vitro and in vivo. *Mol Cell* **23**, 265-71 (2006).
35. Hill, R.J. & Crossan, G.P. DNA cross-link repair safeguards genomic stability during premeiotic germ cell development. *Nat Genet* **51**, 1283-1294 (2019).
36. Kurimoto, K. *et al.* Complex genome-wide transcription dynamics orchestrated by Blimp1 for the specification of the germ cell lineage in mice. *Genes Dev* **22**, 1617-35 (2008).
37. Seisenberger, S. *et al.* The dynamics of genome-wide DNA methylation reprogramming in mouse primordial germ cells. *Mol Cell* **48**, 849-62 (2012).
38. Hajkova, P. *et al.* Epigenetic reprogramming in mouse primordial germ cells. *Mech Dev* **117**, 15-23 (2002).
39. La Salle, S. *et al.* Loss of spermatogonia and wide-spread DNA methylation defects in newborn male mice deficient in DNMT3L. *BMC Dev Biol* **7**, 104 (2007).
40. Hill, P.W.S. *et al.* Epigenetic reprogramming enables the transition from primordial germ cell to gonocyte. *Nature* **555**, 392-396 (2018).
41. Wossidlo, M. *et al.* 5-Hydroxymethylcytosine in the mammalian zygote is linked with epigenetic reprogramming. *Nat Commun* **2**, 241 (2011).
42. Gu, T.P. *et al.* The role of Tet3 DNA dioxygenase in epigenetic reprogramming by oocytes. *Nature* **477**, 606-10 (2011).
43. Hajkova, P. *et al.* Genome-wide reprogramming in the mouse germ line entails the base excision repair pathway. *Science* **329**, 78-82 (2010).
44. Guo, F. *et al.* The Transcriptome and DNA Methylome Landscapes of Human Primordial Germ Cells. *Cell* **161**, 1437-52 (2015).

REVIEWERS' COMMENTS

Reviewer #1 (Remarks to the Author):

The authors have been very thorough in responding to reviewer comments and have addressed all of my concerns. The manuscript has been significantly improved through both added data and revisions to the text. Overall this is a high quality and important new body of work.

One remaining minor comment: In lines 186-187 the authors state "At E12.5, we found that Rev1-/AA embryos had a comparable number of PGCs to wild-type." However, the numbers aren't really comparable, since the Rev1 mutant has 36% fewer PGCs (a difference indicated in the figure to be statistically significant ($p=0.0499$)). This wording therefore should be adjusted, perhaps to say that it is a moderate reduction in PGCs relative to that in Rev1-/ct mutants. The authors emphasize the more severe phenotype in the -/ct mutant to highlight the non-catalytic activities of REV1 but the catalytically dead AA mutant does have fewer PGCs (Fig 2C) as well as fewer follicles and fewer PLZF+ cells (Figs 2E,F), indicating that there is a role for REV1 catalytic activity too.

Reviewer #3 (Remarks to the Author):

NCOMMS-23-27611A:

The authors nicely addressed my previous concerns and comments. I do not have any comments on the scientific results in this paper. However, several issues described below should be resolved before publication for the readers of this paper and the journal.

1. Figure legends are very poor in the description. It is hard to digest the data. For example, there is a poor description of the data.

-The data point is the mean (or median) and what is the error bar (SD) in graphs? What are the statistical methods to calculate the P-values?

-the authors did not explain why they used a marker protein in the assays. For example, in Figure 1d, the authors need to explain what SOX17 and OCT4 are. In Figure 1m, what is SSEA1?

- in Figure 1d, the right panels are enlarged, but it is not mentioned. And it lack the same magnification of the control (WT).

-

2. In Figure 1c, f, g and h, circles of the data are not clearly seen on the background of the bars with the same color. The same is true for other graphs in Figure 3k, l, m, Figure 4, and Supplemental Data Figure 8a, b, d.

3. In Figure 2c, e, and f, Rev1(A/A) shows clear reduced germ cells compared to the control (from P-values). This should be mentioned somewhere either in the text or the legend for the fair evaluation.

4. The graphs of Fig.3i and of Fig.3m (E8.5) look very similar. If so, please clarify. And also the graphs of Fig.3m (E8.5) and Supplemental Data Figure 4g, h, and i look very similar. Please clarify.

5. In Supplemental Data Figure 4e, please compare the statistical significance of survival between two mice. In line 240, the word (similar lifespan) may be too strong if there is a statistical significance.

6. Figure 1d: it would be nice to share the quantification of the immuno-staining analysis.

7. Supplemental Data Figure 5c: please redraw the Y-axis with a range of 0-5.

8. Supplemental Data Figure 7e: It would be nice to have WT control in the graph.

9. Line 511: 5"I" (capital letter) should be 5"i".

Reviewer #1 (Remarks to the Author):

The authors have been very thorough in responding to reviewer comments and have addressed all of my concerns. The manuscript has been significantly improved through both added data and revisions to the text. Overall this is a high quality and important new body of work.

Our response: We are glad that our revisions have satisfied the reviewer and are pleased that they find our work of high quality and importance. We thank the reviewer for improving the manuscript during the process.

One remaining minor comment: In lines 186-187 the authors state "At E12.5, we found that Rev1-/AA embryos had a comparable number of PGCs to wild-type." However, the numbers aren't really comparable, since the Rev1 mutant has 36% fewer PGCs (a difference indicated in the figure to be statistically significant ($p=0.0499$)). This wording therefore should be adjusted, perhaps to say that it is a moderate reduction in PGCs relative to that in Rev1-ct mutants. The authors emphasize the more severe phenotype in the -/ct mutant to highlight the non-catalytic activities of REV1 but the catalytically dead AA mutant does have fewer PGCs (Fig 2C) as well as fewer follicles and fewer PLZF+ cells (Figs 2E,F), indicating that there is a role for REV1 catalytic activity too.

Our response: We thank the reviewer for pointing this out and have now modified the text. The sentence now reads "At E12.5, we found that Rev1^{-AA} embryos had a reduction in the frequency of PGCs however this was moderate when compared to Rev1^{-CT} or Rev1^{-/-} embryos." And at the end of the section "Together these data show that the catalytic activity of REV1 plays a moderate role in germ cell development, however the C-terminus, which coordinates protein-protein interactions during TLS, is critical for PGC development."

Reviewer #3 (Remarks to the Author):

NCOMMS-23-27611A:

The authors nicely addressed my previous concerns and comments. I do not have any comments on the scientific results in this paper. However, several issues described below should be resolved before publication for the readers of this paper and the journal.

Our response: We are glad that our revisions have addressed the reviewer previous concerns. We thank the reviewer for improving the manuscript during the process.

1. Figure legends are very poor in the description. It is hard to digest the data. For example, there is a poor description of the data.

-The data point is the mean (or median) and what is the error bar (SD) in graphs? What are the statistical methods to calculate the P-values?

Our response: We have now included descriptions of what data is presented, the number of observations, and the statistical tests used throughout.

-the authors did not explain why they used a marker protein in the assays. For example, in Figure 1d, the authors need to explain what SOX17 and OCT4 are. In Figure 1m, what is SSEA1?

Our response: We have now included an explanation for the selection of markers in Figure 1d in the results section "the few induced hPGCLCs expressed the canonical germ cell markers SOX17, TFAP2C and OCT4". We have also defined and explained SSEA-1 by adding the following text to the results section "double positive for the marker SSEA-1 (stage specific embryonic antigen-1) and GFP; SSEA1⁺GOF18-GFP⁺"

- in Figure 1d, the right panels are enlarged, but it is not mentioned. And it lack the same magnification of the control (WT).

Our response: We have added a justification for the magnification to the main text results section "The magnified inset shows the nuclear localization of each factor (Fig. 1d)."

2. In Figure 1c, f, g and h, circles of the data are not clearly seen on the background of the bars with the same color. The same is true for other graphs in Figure 3k, l, m, Figure 4, and Supplemental Data Figure 8a, b, d.

Our response: We can change the colour if this is a concern of the production team however, we believe that the points are visible.

3. In Figure 2c, e, and f, Rev1(A/A) shows clear reduced germ cells compared to the control (from P-values). This should be mentioned somewhere either in the text or the legend for the fair evaluation.

Our response: We have modified the text to address this concern and the concern of reviewer 1 (please see above). "Together these data show that the catalytic activity of REV1 plays a moderate role in germ cell development, however the C-terminus, which coordinates protein-protein interactions during TLS, is critical for PGC development."

4. The graphs of Fig.3i and of Fig.3m (E8.5) look very similar. If so, please clarify. And also the graphs of Fig.3m (E8.5) and Supplemental Data Figure 4g, h, and i look very similar. Please clarify.

Our response: We have now clarified in the text that this is the same E8.5 data, it is replotted in Figure 3m to show it on the same axis as the other time points therefore allowing comparison. The data in Supplementary Figure 4g-i have been replotted in 3m again to allow the same axis to be used therefore allowing comparison. We have clarified these points by re-writing the text as follows. "Next, we quantified the number of PGCs by flow cytometry at each day of development between E9.5-12.5, replotting the E8.5 data on the same axis, and found that from E9.5 onwards Rev1^{-/-} and Pcnar/R embryos had a significantly contracted PGC pool when compared to wildtype (Fig. 3m and Supplementary Fig. 4g-i)."

5. In Supplemental Data Figure 4e, please compare the statistical significance of survival between two mice. In line 240, the word (similar lifespan) may be too strong if there is a statistical significance.

Our response: We have now tested this significance of this using the Mantel-Cox test giving $p=0.6013$. As there is no significant difference, we have left the wording unchanged.

6. Figure 1d: it would be nice to share the quantification of the immuno-staining analysis.

Our response: We quantified the frequency of hPGCLCs in by flow cytometry as this allows us to sample many more cells than would be possible by immunofluorescence, this quantitative data is shown in Fig 1b and 1c.

7. Supplemental Data Figure 5c: please redraw the Y-axis with a range of 0-5.

Our response: We have redrawn the axis as requested and the statistical p-values have been added to the legend due to constraints on the space.

8. Supplemental Data Figure 7e: It would be nice to have WT control in the graph.

Our response: The data showing the change in wildtype ovary mass over time is shown in Supplementary Fig. 7d. In Figure 7e we are comparing the different TLS mutants and feel that adding the data will crowd the figure.

9. Line 511: 5"l" (capital letter) should be 5"l".

Our response: We thank the reviewer for pointing out this error which we have now corrected.